# LAW OF THE WEAKEST LINK:
# CROSS CAPABILITIES OF LARGE LANGUAGE MODELS

**Ming Zhong**[*1,2]  **Aston Zhang**[*1]  **Xuewei Wang**[1]  **Rui Hou**[1]  **Wenhan Xiong**[1]  **Chenguang Zhu**[1]
**Zhengxing Chen**[1]  **Liang Tan**[1]  **Chloe Bi**[1]  **Mike Lewis**[1]  **Sravya Popuri**[1]  **Sharan Narang**[1]
**Melanie Kambadur**[1]  **Dhruv Mahajan**[1]  **Sergey Edunov**[1]  **Jiawei Han**[2]  **Laurens van der Maaten**[1]
[1]Llama Team, AI @ Meta   [2]University of Illinois Urbana-Champaign

## ABSTRACT

The development and evaluation of Large Language Models (LLMs) have largely focused on individual capabilities. However, this overlooks the intersection of multiple abilities across different types of expertise that are often required for real-world tasks, which we term **cross capabilities**. To systematically explore this concept, we first define seven core individual capabilities and then pair them to form seven common cross capabilities, each supported by a manually constructed taxonomy. Building on these definitions, we introduce CROSSEVAL, a benchmark comprising 1,400 human-annotated prompts, with 100 prompts for each individual and cross capability. To ensure reliable evaluation, we involve expert annotators to assess 4,200 model responses, gathering 8,400 human ratings with detailed explanations to serve as reference examples. Our findings reveal that current LLMs consistently exhibit the "Law of the Weakest Link," where cross-capability performance is significantly constrained by the weakest component. Across 58 cross-capability scores from 17 models, 38 scores are lower than all individual capabilities, while 20 fall between strong and weak, but closer to the weaker ability. These results highlight LLMs' underperformance in cross-capability tasks, emphasizing the need to identify and improve their weakest capabilities as a key research priority. The code, benchmarks, and evaluations are available on our project website.

## 1 INTRODUCTION

The development and evaluation of Large Language Models (LLMs) (OpenAI, 2023; 2024; Anthropic, 2024; Reid et al., 2024) have predominantly centered on individual capabilities. Developers commonly construct specialized datasets tailored to distinct abilities, and then train models by blending these data sources. For instance, Llama 3's post-training incorporates a mix of data, from general English to code and multilingual content, among others, each subset aimed at honing a specific skill (Llama Team, 2024). Evaluation methods follow a similar pattern, with benchmarks typically assessing these abilities in isolation, offering a snapshot of how well a model can reason (Clark et al., 2018; Cobbe et al., 2021; Hendrycks et al., 2021b), code (Chen et al., 2021; Austin et al., 2021), or manage factual knowledge (Hendrycks et al., 2021a).

However, can all real-world tasks be adequately categorized under just one capability, or do they frequently demand the seamless integration of multiple skills, thereby challenging the prevalent approach to evaluating these advanced LLMs? Consider a user prompt asking, "Which direction has the total rainfall in Tokyo, Japan been trending over the past 10 years? Explain it step by step." Such a task requires the integration of tool use (web browsing) with analytical reasoning. Similarly, when a developer provides HTML and JavaScript for an API-driven application and asks, "Give me a basic understanding of what this web app does," the model must combine long-context comprehension with coding expertise. We define these scenarios as **cross capabilities**—the intersection of multiple distinct capabilities across different types of expertise necessary to address complex, real-world tasks. This discrepancy between the isolated focus of current LLM evaluation and the multifaceted demands of user interactions raises a critical question:

*How does the performance of LLMs on tasks requiring cross capabilities reflect or diverge from their performance in individual capabilities?*

This question opens up various possibilities for portraying the relationship between distinct abilities in LLMs and their collective performance. Insights from multiple fields can shed light on these dynamics. For example, "Synergy Theory" (Corning, 1983) suggests that the interaction of different components in a system can produce effects greater than the sum of individual parts, while "Compensatory Mechanism" (Adler, 1917), a concept from psychology, introduces that stronger abilities within a system can offset weaker ones. Additionally, "Law of the Weakest Link" (Liebig, 1840) presents that a system's performance is limited by its weakest element, and the idea of "Emergent Properties" (Anderson, 1972) highlights how new behaviors can arise from the interaction of components, which are not predictable from their individual components alone. Given the substantial investment in enhancing the particular abilities of LLMs, identifying how individual capabilities impact performance on tasks requiring cross abilities is crucial for guiding future development.

In this paper, we investigate how the interplay of individual capabilities influences collective performance, with the goal of providing insights for advancing LLM effectiveness in handling cross-capability tasks. Specifically, our research explores the following key questions:

- **RQ1: How can we comprehensively define individual and cross capabilities in LLMs?** To effectively define all capabilities in LLMs, we must systematically categorize tasks that reflect real-world interactions. We identify seven core individual capabilities, including *English*, *Reasoning*, *Coding*, *Image Recognition*, *Tool Use*, *Long Context*, and *Spanish*, and pair them to form seven common cross capabilities, such as *Coding & Reasoning*. For each capability, we provide clear definitions and manually construct a detailed taxonomy that links the capability to complex tasks, systematically breaking it down into two hierarchical levels: broad categories at the first level and specific tasks at the second. These taxonomies lay the groundwork for constructing benchmarks that can comprehensively cover and assess a broader range of LLM capabilities.

- **RQ2: How can we benchmark both individual and cross capabilities in LLMs?** To benchmark all capabilities in LLMs, we construct a detailed evaluation framework, CROSSEVAL, based on manually annotated prompts that align with our established taxonomy. Each prompt is categorized by capability and difficulty, ensuring thorough coverage of both individual and cross capabilities. We collect multiple model responses for each prompt and engage expert human annotators to rate and explain these responses. In total, CROSSEVAL comprises 1,400 prompts, 4,200 model responses, and 8,400 expert ratings with detailed explanations. Finally, we introduce LLM-based evaluators to assess responses using these reference examples, achieving strong agreement with human judgments, thereby establishing a reliable benchmark for evaluating LLM performance across a wide spectrum of open-ended tasks.

- **RQ3: What patterns exist in the relationship between individual and cross-capability performance in LLMs?** Through extensive evaluation using CROSSEVAL, we uncover clear patterns in the relationship between individual and cross-capability performance. Most notably, cross-capability performance is typically constrained by the weakest capability, following the "Law of the Weakest Link" effect. This pattern is consistent across different LLMs and evaluators, suggesting that deficiencies in an individual capability can significantly limit overall performance in more complex tasks. Specifically, of the 58 cross-capability scores from 17 models, 38 fall below the individual capabilities, while 20 lie between the strong and weak, skewing towards the weaker. These results underscore the need for targeted optimization to strengthen the weakest capabilities, especially in areas like *Tool Use*, where models struggle the most.

- **RQ4: How do shifts in individual capabilities impact cross-capability performance in LLMs?** Beyond evaluating the static relationship between individual and cross capabilities, we investigate how altering individual capabilities affects cross-capability performance. Through case studies using a principle-based system prompting method, we selectively enhance specific capabilities and find that improvements in weaker capabilities lead to substantial gains in cross-capability tasks, while changes in stronger capabilities often result in only minor shifts. This finding further corroborates the "Law of the Weakest Link" effect, as an LLM's cross-capability performance continues to conform to this phenomenon when its individual capability performance changes.

In summary, this paper highlights the critical oversight of cross capabilities in LLMs, despite being essential for real-world tasks. To systematically explore it, we establish a comprehensive benchmark to model both individual and cross capabilities, revealing the "Law of the Weakest Link" effect. Given that LLMs generally underperform in cross-capability tasks, identifying and enhancing these weak points should be a priority for future research and development.

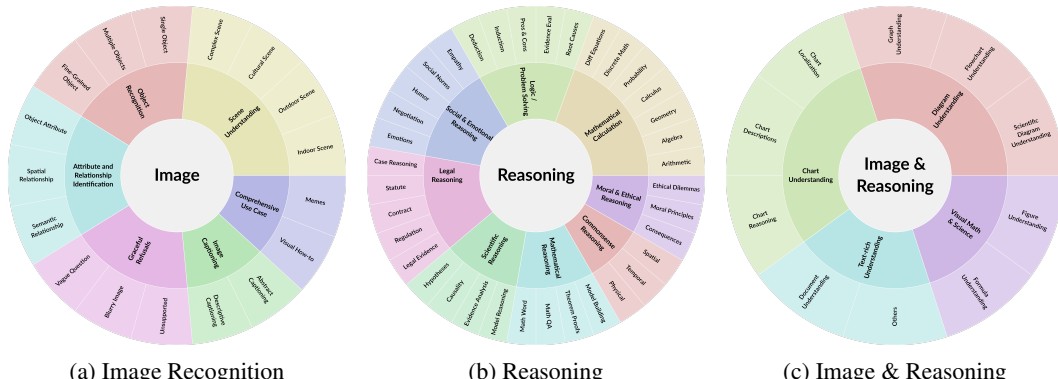

| (a) Image Recognition | (b) Reasoning | (c) Image & Reasoning |

Figure 1: **Taxonomy visualizations for Image Recognition, Reasoning, and the corresponding cross capability.** Each node represents a specific type of task. The first two taxonomies illustrate tasks that require only individual capabilities for LLMs to complete. The final taxonomy, however, depicts tasks that lie at the intersection of *Image Recognition* and *Reasoning* capabilities, necessitating the use of both abilities to accomplish them. For the full taxonomy of all the individual and capabilities and cross capabilities, please see Appendix B.

## 2 DEFINING INDIVIDUAL & CROSS CAPABILITIES IN LLMS

Real-world interactions with LLMs encompass tasks that may require either an individual capability or the simultaneous engagement of distinct skills. To evaluate LLMs effectively, defining and differentiating these capabilities is crucial. In this section, we identify seven individual and seven cross capabilities that reflect a broad spectrum of user queries and organize them into taxonomies.

As illustrated in Figure 1, the taxonomies follow a hierarchical structure: the root represents an individual or cross capability, while the next two layers (Level-1 and Level-2) break them into more specific tasks. This framework helps distinguish tasks based on single or multiple capabilities, enabling comprehensive LLM evaluations. We next detail the selected capabilities.

### 2.1 INDIVIDUAL CAPABILITIES

We begin by selecting seven core individual capabilities of LLMs: *English*, *Reasoning*, *Coding*, *Image Recognition*, *Tool Use*, *Long Context*, and one representative of multilingual capabilities, *Spanish*. Each of these is defined as follows:

- **English:** The ability to understand, generate, and manipulate English text for tasks such as factual answering, procedural instructions, creative writing, and dialogue.
- **Reasoning:** The ability to perform logical deduction, mathematical computation, analytical problem-solving, and domain-specific reasoning, including areas such as scientific, legal, and ethical contexts.
- **Coding:** The ability to write, synthesize, document, debug, and review code across various programming languages.
- **Image Recognition:** The ability to perceive, identify, interpret, and describe visual information.
- **Tool Use:** The ability to interact with external tools and APIs, including functions such as web browsing, code execution, and file uploads.
- **Long Context:** The ability to process, comprehend, and synthesize information to generate responses from extensive textual inputs, ranging from 16K to 128K tokens.
- **Spanish:** The same as the English capability, but for the Spanish language.

The selection of individual capabilities is primarily based on the capability evaluations outlined for post-trained LLMs in Llama 3 (Llama Team, 2024), as these represent key areas of interest within the LLM community. Additionally, we include *Image Recognition* to account for the expanding multimodal capabilities of state-of-the-art LLMs, ensuring that our findings are applicable across both language and vision modalities. Appendix B.1 provides the full taxonomy of individual capabilities.

## 2.2 CROSS CAPABILITIES

We explore cross-capability scenarios by combining two capabilities. To achieve this, we pair the individual capabilities described earlier and select seven common combinations: *Coding & Reasoning*, *Image Recognition & Reasoning*, *Tool Use & Coding*, *Tool Use & Reasoning*, *Long Context & Coding*, *Spanish & Reasoning*, and *Spanish & Image Recognition*. Below is the Level-1 taxonomy:

- **Coding & Reasoning:** Coding Q&A (Text to Text) (5), Code Explanation (2), Programming Assistant (5), Mathematical Calculation (7).

- **Image Recognition & Reasoning:** Diagram Understanding (3), Chart Understanding (3), Text-Rich Understanding (2), and Visual Math and Science (2).

- **Tool Use & Coding:** Code Execution (3), Code Debugging with Execution (2), Programming Assistant with Execution (1), and Code Execution with File Uploads (3)

- **Tool Use & Reasoning:** Mathematical Reasoning (2), Scientific Reasoning (15), and Mathematical Calculation (13).

- **Long Context & Coding:** Repository-Level Code Generation (5), Repository-Level Code Understanding (2), Repository-Level Code Debugging (1), Log Analysis (3), and API Docs Understanding (2).

To clarify, the number in parentheses represents the Level-2 subcategories within each Level-1 category. To select cross-capability scenarios, we ensure comprehensive coverage of all individual capabilities and analyze the frequency of capability pairs using data from public real user queries (Zheng et al., 2024; Lin et al., 2024). This analysis enables us to identify the seven most common and practically relevant pairs. The full taxonomy of cross capabilities is available in Appendix B.2.

## 3 CROSSEVAL BENCHMARK CONSTRUCTION

This section outlines the annotation process for building the CROSSEVAL benchmark and the configuration of the LLM used as its evaluator.

### 3.1 PROMPT SET ANNOTATION

The prompt set forms the foundation for benchmarking LLMs, but real-world prompts often contain low-quality inputs, making it hard to distinguish advanced models (Li et al., 2024). Additionally, creating sufficiently difficult prompts is inherently challenging (Padlewski et al., 2024). To address this, we adopt a comprehensive annotation process to ensure both quality and difficulty levels.

**Annotation Procedure.** In this paper, we focus on single-turn, open-ended prompts. The annotation process follows a "category-first, prompt-second" approach to ensure comprehensive coverage of our taxonomy. Specifically, annotators from the data vendor are instructed to randomly select a leaf node from the taxonomy, which determines the category to be annotated. Once the category is selected, annotators create prompts aligned with the provided guidelines. Each capability has clear criteria for three difficulty levels—easy, medium, and hard—detailed in Appendix D.6. For *Spanish* as an individual capability, all prompts are annotated from scratch, without overlap with the *English* set. In cross-capability scenarios involving *Spanish*, the prompt sets are derived by translating the associated English-based prompts. For instance, the *Spanish & Reasoning* set is created by translating the *Reasoning* prompts into Spanish.

To maintain consistency and high quality, we begin with a pilot annotation phase where the authors act as reviewers, providing feedback to identify any issues with the initial annotations and refine the annotation guidelines accordingly. Afterward, the main annotation phase begins, resulting in 100 to 500 prompts for each capability, depending on the size of the annotator pool assigned to it. Reviewers then perform quality checks and apply filtering to produce a final set of 100 high-quality prompts per capability. This process ensures the difficulty distribution follows the standards used in Llama 3's human evaluations, with 10% easy, 30% medium, and 60% hard prompts (Llama Team, 2024). Ultimately, the final prompt set consists of 1,400 prompts, with 100 prompts for each capability, covering all 76 Level-1 and 332 Level-2 categories as listed in Section §2.

## 3.2 Multiple References with Human Annotations

Providing a gold reference for each instance, while standard before the rise of LLMs, is not feasible for our prompt set for three reasons: 1) Open-ended queries often lack a single correct answer, and relying on one reference risks introducing evaluation bias. 2) Prompts requiring domain expertise, such as coding or mathematics, are sometimes too difficult even for college-level expert annotators. 3) For tool-use prompts, like "What is the temperature in the Bay Area today?," the correct response is dynamic and changes over time. To address this, we propose using multiple model responses, scored and explained by human annotators, to serve as references for evaluation.

**Annotator Qualifications.** For all annotations in this paper, we use the same data vendor, employing professional experts with domain-specific knowledge, such as reasoning, coding, and Spanish. The data vendor selects the appropriate annotator pool based on the capabilities being evaluated. While creating a definitive gold reference is impractical, our annotators are capable of assessing the correctness of model responses and providing well-justified ratings.

**Model Response Collection.** For each prompt, we aim to gather three distinct model responses representing varying levels of quality: low, medium, and high. These responses are randomly drawn from various models within the Llama and GPT model families, including Llama 3.1 8B/70B/405B and different versions of GPT-4. For capabilities involving *Reasoning*, *Image Recognition*, and *Tool Use*, we manually annotate one response if all three collected responses contain noticeable errors.

**Annotating Human Ratings with Explanations.** For each model response, two independent annotators rate it on a 1–5 Likert scale, providing a paragraph to explain their rating. Multiple reference examples are offered in Appendix D.5. We track inter-rater agreement and find that evaluating model responses can be challenging, even for expert annotators, making consensus difficult to achieve.

To enhance consistency, we initially annotate 30% of the prompt set in a pilot phase. During this phase, the inter-rater agreement is 33.65%, with a Krippendorff's Alpha (K-Alpha) (Krippendorff, 2018) of 0.48, indicating relatively poor agreement. We then conduct the second and third rounds of annotation, allowing new raters from the same pool to review previous annotations, better understand the scoring criteria, and provide their ratings with explanations. After each round, we update the guidelines to improve the annotation process. This iterative procedure proves effective: inter-rater agreement improves from 33.65% to 45.79%, and finally to 47.38%, while K-Alpha increases from 0.48 to 0.66, and eventually to 0.73. After completing these rounds, we apply the updated guidelines to annotate the full dataset using the same trained annotator pool. On the full dataset, the inter-rater agreement rate reaches 54.93%, with a K-Alpha of 0.76.

For comparison, in Chatbot Arena (Zheng et al., 2023), the human agreement rate is 81% for binary classification (win/lose) and 63% for a 1–3 scale (win/tie/lose). In contrast, we independently score each response on a more granular 1-5 scale, yet still achieve a substantial level of agreement.

**CrossEval Benchmark Statistics.** The final CROSSEVAL benchmark comprises 1,400 prompts across 14 capabilities, 4,200 reference model responses, and 8,400 human ratings with accompanying explanations. Additionally, we provide several examples of the prompt set, along with human ratings and explanations, in Appendix D.4 and D.5, respectively.

## 3.3 Building LLM-based Evaluators

CROSSEVAL is also the largest benchmark available for measuring the correlation between LLM rating and human judgments. With each prompt containing three responses and six human ratings, it enables the exploration of the most effective in-domain LLM evaluators for this benchmark.

### 3.3.1 Prompting LLMs for Evaluation

While the LLM-as-a-Judge paradigm has gained popularity (Zheng et al., 2023), there is no standardized method for designing prompts or for guiding LLMs to output evaluation scores. Common practices include generating an answer first, setting evaluation rules manually, and then instructing the model to assign a score to the response being evaluated (Zeng et al., 2024).

In practice, we find that self-generated answers frequently lead to issues. For instance, response length can exceed model limits, preventing the model from generating a score. This approach also

| Capabilities | GPT-4o mini | Llama 3.1 405B | Claude 3.5 Sonnet | GPT-4o-05-13 |
|---|---|---|---|---|
| English | 0.383 | 0.452 | **0.516** | 0.498 |
| Reasoning | 0.681 | 0.699 | 0.704 | **0.731** |
| Coding | **0.627** | 0.568 | 0.599 | 0.624 |
| Image Recognition | 0.576 | – | 0.733 | **0.760** |
| Tool Use | 0.587 | 0.609 | **0.683** | 0.629 |
| Long Context | 0.405 | 0.500 | **0.609** | 0.594 |
| Spanish | 0.552 | 0.536 | **0.596** | 0.594 |
| Coding & Reasoning | 0.618 | 0.600 | 0.623 | **0.664** |
| Image & Reasoning | 0.701 | – | **0.819** | 0.775 |
| Tool Use & Coding | 0.484 | 0.545 | 0.588 | **0.639** |
| Tool Use & Reasoning | 0.642 | 0.698 | 0.665 | **0.729** |
| Long Context & Coding | 0.524 | 0.535 | **0.620** | 0.593 |
| Spanish & Reasoning | 0.691 | 0.734 | 0.715 | **0.772** |
| Spanish & Image | 0.556 | – | **0.752** | 0.669 |
| Overall Pearson ($r$) | 0.621 | – | 0.696 | **0.697** |
| Overall Spearman ($r_s$) | 0.609 | – | 0.676 | **0.679** |
| Overall Kendall ($\tau$) | 0.508 | – | 0.550 | **0.560** |

Table 1: **Correlations between LLM ratings and human judgments.** The top section shows Pearson correlations across individual and cross capabilities for four LLMs, and the bottom three shaded rows present the overall correlations.

causes the LLMs to overly rely on their own generated answers, overlooking valuable insights from human-annotated references. To solve this, we propose the following prompting strategy:

**Multi-References-based Prompting.** We provide the 1-5 Likert scale rubrics in the system prompt (see Table 34 in Appendix D.7), followed by any relevant attachments (e.g., an image) and the user prompt. When evaluating LLM-as-a-Judge performance, up to two reference responses with scores and explanations can be included. For example, when evaluating a medium-quality response, we provide low- and high-quality responses with their ratings for context. For new model evaluations, all three model responses are included with human annotations as references.

**Point Deduction-based Prompting.** As noted in prior studies (Zheng et al., 2023), LLM-as-a-Judge often favors longer, more structured responses, leading to inflated evaluation scores. To mitigate this, we no longer have LLMs directly generate their own answers and assign scores. Instead, they summarize issues in both the reference examples and the evaluated response, specifying point deductions (Zhong et al., 2024). This approach ensures more balanced and systematic evaluations. The complete prompt is detailed in Table 35 in Appendix D.7.

### 3.3.2 CORRELATIONS WITH HUMAN JUDGEMENTS

To assess the effectiveness of the LLM evaluator on CROSSEVAL, we conduct experiments with four advanced LLMs: GPT-4o mini, Llama 3.1 405B (Llama Team, 2024), Claude 3.5 Sonnet (Anthropic, 2024), and GPT-4o (OpenAI, 2023). For each prompt, we provide two reference examples and ask the LLM to evaluate the third, comparing model-based scores with the average human rating. We evaluate 4,200 samples across 14 capabilities, with correlations shown in Table 1.

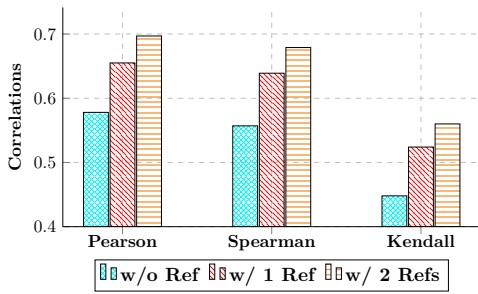

Figure 2: **Ablation study on the number of reference examples.**

Each LLM demonstrates strengths in different areas. For example, Claude 3.5 Sonnet excels in evaluating *Tool Use*, *Image Recognition & Reasoning*, and *Spanish & Image Recognition*, while GPT-4o performs best in cross capabilities like *Coding & Reasoning*, *Tool Use & Coding*, and *Spanish & Reasoning*. For comparison, the recent BigGen Bench (Kim et al., 2024) reports a Pearson correlation of 0.627 with gold references, while our score approaches 0.7, showing that multiple reference examples still enable reliable evaluations without a gold reference. GPT-4o achieves the highest correlations overall, so we select it as the final evaluator, with results from Claude in Appendix E.2 for reference. Further details and discussions on evaluator selection are available in Appendix D.8.

| Individual Capabilities | | | | | | |
|---|---|---|---|---|---|---|
| **Models** | **English** | **Reasoning** | **Coding** | **Image** | **Tool Use** | **Long Context** | **Spanish** |
| GPT-4o mini | 73.64 | 69.31 | 71.17 | 65.23 | – | 76.18 | 74.51 |
| GPT-4o | 76.12 | 72.84 | 72.03 | 73.02 | – | 77.17 | 78.10 |
| o1-mini | 75.25 | 81.02 | 80.70 | – | – | 76.74 | 79.09 |
| o1-preview | 78.59 | 82.30 | 79.09 | – | – | 78.90 | 79.64 |
| Claude 3 Haiku | 63.87 | 56.81 | 61.64 | 51.00 | – | 69.68 | 67.95 |
| Claude 3 Sonnet | 69.19 | 62.88 | 66.09 | 56.56 | – | 72.40 | 69.43 |
| Claude 3 Opus | 68.94 | 66.22 | 69.68 | 61.76 | – | 74.69 | 74.01 |
| Claude 3.5 Sonnet | 75.00 | 71.54 | **74.01** | 68.57 | – | 74.32 | 76.12 |
| Gemini 1.5 Flash | 66.59 | 63.25 | 65.60 | 56.81 | – | 73.52 | 70.05 |
| Gemini 1.5 Pro | 71.91 | 70.61 | 69.56 | 69.56 | – | **76.51** | 74.26 |
| Gemini 1.5 Pro Exp | **75.87** | **73.02** | 69.56 | **71.17** | – | 75.37 | **76.24** |
| Reka Edge | 52.23 | 45.30 | 39.36 | 48.89 | – | 37.01 | 52.48 |
| Reka Flash | 63.87 | 62.63 | 57.68 | 56.38 | – | 55.82 | 68.07 |
| Reka Core | 71.54 | 68.69 | 62.38 | 56.94 | – | 60.90 | 73.77 |
| Llama 3.1 8B | 64.11 | 53.97 | 55.08 | – | 42.09 | 59.53 | 55.70 |
| Llama 3.1 70B | 68.82 | 62.88 | 65.47 | – | 47.04 | 68.82 | 64.48 |
| Llama 3.1 405B | 73.52 | 69.31 | 69.19 | – | **47.90** | 69.31 | 72.59 |
| Cross Capabilities | | | | | | |
| **Models** | **Coding & Rea.** | **Image & Rea.** | **Long & Coding** | **Spanish & Rea.** | **Spanish & Image** | **Tool & Coding** | **Tool & Rea.** |
| GPT-4o mini | 72.03 | 65.60 | 65.10 | 69.56 | 65.10 | – | – |
| GPT-4o | 73.33 | 71.29 | 67.95 | 73.52 | 74.63 | 45.80 | 54.41 |
| o1-mini | 79.21 | – | 76.12 | 79.83 | – | – | – |
| o1-preview | 79.58 | – | 73.39 | 80.70 | – | – | – |
| Claude 3 Haiku | 58.05 | 49.88 | 58.67 | 57.80 | 52.85 | – | – |
| Claude 3 Sonnet | 61.14 | 54.71 | 58.79 | 60.77 | 60.52 | – | – |
| Claude 3 Opus | 63.37 | 53.84 | 58.17 | 67.33 | 64.11 | – | – |
| Claude 3.5 Sonnet | **71.41** | **69.43** | 65.72 | 70.55 | 69.81 | – | – |
| Gemini 1.5 Flash | 64.73 | 51.74 | 62.13 | 65.10 | 53.10 | – | – |
| Gemini 1.5 Pro | 69.68 | 67.95 | **65.97** | 69.56 | 62.26 | – | – |
| Gemini 1.5 Pro Exp | 67.33 | 69.06 | **65.97** | 71.54 | 70.18 | – | – |
| Reka Edge | 41.34 | 28.60 | 20.43 | 40.97 | 45.06 | – | – |
| Reka Flash | 56.94 | 43.45 | 37.63 | 59.66 | 55.82 | – | – |
| Reka Core | 63.62 | 46.66 | 41.25 | 68.01 | 54.71 | – | – |
| Llama 3.1 8B | 55.08 | – | 45.06 | 46.42 | – | 46.91 | 43.82 |
| Llama 3.1 70B | 67.21 | – | 50.50 | 59.41 | – | 50.25 | 49.45 |
| Llama 3.1 405B | 66.96 | – | 54.58 | 64.48 | – | **52.23** | **51.74** |

Table 2: **Experimental results on the CROSSEVAL benchmark.** In cross-capability evaluations, we define one of the involved individual capabilities as stronger and the other as weaker if the absolute score difference between them exceeds $\Delta = 3$ points. In 58 cross-capability scenarios where this difference is present (indicated by a colored background), 38 cases show performance lower than both individual capabilities ( red background ), and 20 show performance between the two but closer to the weaker capability ( blue background ). Notably, no cross-capability score ever comes close to or exceeds the stronger individual capability.

**Discussion on Reference Examples.** Given the substantial effort invested in collecting and annotating reference examples, ensuring their effectiveness for evaluation is crucial. To this end, we conduct ablation studies with GPT-4o to assess how the number of reference examples impacts the correlation metrics, as illustrated in Figure 2. A clear trend emerges: as the number of reference examples increases, all three correlation metrics improve significantly. For example, the Pearson correlation starts at 0.578 with no reference examples, rises to 0.655 with one reference, and reaches 0.697 with two references. Notably, when evaluating new responses on CROSSEVAL, we include all three reference examples, potentially achieving even higher correlations for more accurate evaluations.

## 4 "LAW OF THE WEAKEST LINK" OF CROSS CAPABILITIES IN LLMS

In this section, we explore the relationship between individual and cross capabilities in LLMs. We first present the experimental setup, followed by a detailed discussion of the findings.

### 4.1 FINDINGS ON THE CROSSEVAL BENCHMARK

To better present the results, we linearly map the average scores for each capability from a 1–5 scale to a 1–100 scale. The full results of 17 LLMs are provided in Table 2. Since LLMs tend to prefer self-generated answers (Zheng et al., 2023), we exclude GPT's results from the comparative analysis and treat them as a reference point. Our experiments reveal several key findings:

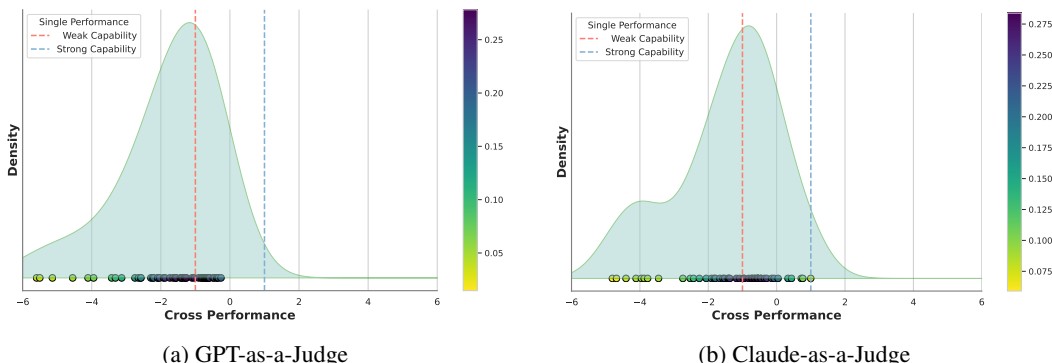

(a) GPT-as-a-Judge                    (b) Claude-as-a-Judge

Figure 3: **Density distribution of cross-capability performance compared to the two individual capabilities.** The plot illustrates a pronounced "Law of the Weakest Link" effect in LLMs, where performance in cross-capability tasks tends to cluster around the weaker individual capability.

**CROSSEVAL effectively differentiates advanced models.** The CROSSEVAL benchmark successfully distinguishes between state-of-the-art LLMs. For example, the four Claude model variants achieve progressively higher reasoning scores: 56.81, 62.88, 66.22, and 71.54, reflecting the increasing capabilities of larger parameter models (Haiku $\Rightarrow$ Sonnet $\Rightarrow$ Opus) or updated versions (Claude 3 $\Rightarrow$ Claude 3.5). Similar trends are observed across all model families, demonstrating that CROSSEVAL is capable of capturing subtle performance differences across capabilities.

**LLMs exhibit a "Law of the Weakest Link" effect in cross capabilities.** To better understand how individual and cross capabilities interact, we identify "strong" and "weak" capabilities within cross-capability tasks when the absolute difference between their individual scores exceeds $\Delta = 3$. In all cases where a distinct strong and weak capability is present, cross-capability performance either matches or slightly underperforms the weaker capability. This suggests that performance on tasks requiring multiple abilities is constrained by the weakest component, a phenomenon closely aligned with the "Law of the Weakest Link" (Liebig, 1840). Much like a barrel's capacity is limited by its shortest stave, the weakest capability in LLMs dictates overall performance.

**"Law of the Weakest Link" is evaluator-agnostic.** To further validate this effect, we normalize strong and weak capability scores to a standardized scale from -1 to 1 and plot the density of cross-capability performance relative to these scores. A score below -1 indicates that the cross-capability performance falls below the weaker individual capability, while 0 represents the average of the two. As shown in Figure 3, the "Law of the Weakest Link" holds true regardless of the evaluator used. With GPT, the density peaks slightly below the weaker capability, while Claude peaks slightly above it. However, in both cases, performance clusters closely around the weaker capability. Moreover, we explore varying $\Delta$ values in E.3, consistently confirming the "Law of the Weakest Link."

Given that many real-world tasks require integrating multiple capabilities, this finding offers key insights for future LLM development. The "Law of the Weakest Link" suggests that deficiencies in one capability can substantially limit performance across any tasks involving that capability. CROSSEVAL provides a foundation for identifying LLM weaknesses, but further research is needed to more comprehensively diagnose and address these deficiencies without compromising other capabilities.

**Tool Use is currently the most challenging capability for LLMs.** Among the capabilities tested, *Tool Use* stands out as the most challenging. Our prompt set includes tasks involving web browsing and code interpretation, and Llama 3.1 is the only model family that currently supports both. However, even Llama 3.1 405B struggled with *Tool Use*, scoring below 50 on this individual capability and only slightly above 50 on tasks combining *Tool Use* with *Coding* or *Reasoning*. These scores are significantly lower than those for other capabilities, indicating a critical area for improvement.

**LLMs underperform in cross-capability tasks.** Despite our efforts to maintain a consistent difficulty level across both individual and cross-capability tasks, LLMs generally perform worse on tasks requiring multiple capabilities. For instance, in the *Spanish & Reasoning* and *Spanish & Image Recognition* tasks, where prompts are direct translations from their English counterparts, the models underperform in most cases compared to individual capabilities. Across all models, the average score for individual capabilities is 65.72, compared to 58.67 for cross capabilities, reveal-

| Models | Individual Capabilities | | | Cross Capabilities | | |
|---|---|---|---|---|---|---|
| | Reasoning | Image Recognition | Spanish | Image & Rea. | Spanish & Rea. | Spanish & Image |
| Claude 3 Haiku | 56.81 | 51.00 | 67.95 | 49.88 | 57.80 | 52.85 |
| + Reasoning | **59.66** | 50.01 | **68.20** | 46.42 | **59.04** | 52.11 |
| + Image | 55.45 | **54.71** | 64.98 | **54.46** | 57.55 | **55.08** |
| + Spanish | 55.20 | 53.59 | 67.21 | 50.13 | 56.81 | 53.72 |
| Gemini 1.5 Flash | 63.25 | 56.81 | 70.05 | 51.74 | 65.10 | 53.10 |
| + Reasoning | **66.71** | 62.50 | **71.29** | **54.46** | **66.59** | 59.04 |
| + Image | 59.91 | **63.00** | 69.43 | 51.61 | 62.13 | **61.76** |
| + Spanish | 61.39 | 61.89 | 69.06 | 52.60 | 64.86 | 58.42 |

Table 3: **Investigating the impact of individual-capability alterations on cross-capability performance.** "+ X" indicates the application of principle-based system prompting to enhance the specific capability X. Results demonstrate that enhancing weaker capabilities leads to more substantial improvements in cross-capability performance compared to enhancing stronger capabilities.

ing a significant performance gap. This disparity demonstrates that current LLMs remain heavily optimized for individual capabilities, with a limited focus on cross-capabilities performance.

# 5 INVESTIGATING INDIVIDUAL-CAPABILITY ALTERATIONS

Beyond evaluating the static capabilities of LLMs, we explore the crucial follow-up question in this section: *how altering specific capabilities affects overall cross-capability performance?*

## 5.1 PRINCIPLE-BASED SYSTEM PROMPTING

To explore the impact of altering individual capabilities without significantly affecting others, we propose a principle-based method that iteratively refines the system prompt to selectively boost specific capabilities of LLMs, based on the responses and evaluations on CROSSEVAL. This approach allows for controlled investigation into cross-capability performance dynamics.

Our method iteratively refines the prompt through operations such as adding, replacing, revising, or keeping principles. After 100 iterations, GPT-4o generates a tailored principle-based prompt to guide the LLM in prioritizing key performance aspects like format adherence, problem-solving strategies, or error avoidance. Detailed prompts used are available in Table 41 in Appendix F.1.

## 5.2 CASE STUDY FOR INVESTIGATION

Experimentally, we select three tasks with the largest gaps relative to individual performance: *Image Recognition & Reasoning*, *Spanish & Reasoning*, and *Spanish & Image Recognition*. We also focus on two LLMs, Claude 3 Haiku and Gemini 1.5 Flash, which display the most significant discrepancies in these tasks. These combinations are chosen because the more pronounced gaps provide clearer insights into the impact of selective capability enhancement on overall performance. Table 3 shows the full results, leading to the following key observations:

**Principle-based system prompting is particularly effective in enhancing weaker capabilities.** For *Reasoning* capability, both models show substantial improvements: Claude 3 Haiku increases by 2.85 points, and Gemini 1.5 Flash improves by 3.46 points. The improvements are even more pronounced in *Image Recognition*, with Claude 3 Haiku improving by 3.71 points and Gemini 1.5 Flash by 6.19 points. These results indicate that the principles derived from the CROSSEVAL evaluation process offer effective guidance for enhancing weaker LLM capabilities, even when applied solely as system prompts. However, for stronger capabilities like *Spanish*, the same method shows limited efficacy, suggesting that refining already-strong capabilities is more challenging.

**"Law of the Weakest Link" effect persists after individual-capability alterations.** Our case study also confirms that performance shifts in individual capabilities continue to conform to the "Law of the Weakest Link" effect. Specifically, altering the weaker capability in a cross-capability scenario has a significant effect on overall performance, while changes to the stronger capability result in only minor adjustments. For example, in the *Image Recognition & Reasoning* scenario with Claude 3 Haiku, when we introduce a system prompt focused on reasoning, the stronger capability (*Reasoning*) improves by 2.85 points, but the weaker capability (*Image Recognition*) drops

by 0.99 points, leading to an overall performance decrease of 3.46 points. Conversely, when an image-related system prompt is added, the weaker capability improves by 3.71 points, the stronger capability decreases by 1.36 points, but the cross-capability performance increases by 4.58 points.

In 10 out of the 18 cross-capability scores examined across the two models, we observe one individual capability improving while the other declines. Notably, in 90% of these cases, changes in cross-capability performance closely follow the trends of the weaker capability. This strong alignment with the "Law of the Weakest Link" underscores the importance of addressing the weakest links in LLM capabilities to drive meaningful improvements in complex, real-world tasks.

**Conclusion of the case study.** These case studies offer further insights into how LLMs conform to the "Law of the Weakest Link". We show that targeted enhancement of weaker capabilities results in more significant improvements in cross-capability performance than focusing on stronger capabilities. Since LLMs underperform in cross-capability tasks, prioritizing the identification and enhancement of the weakest points should be a key focus for future research and development.

## 6 RELATED WORK

**Evaluation of LLMs.** The advancements in LLMs have shifted the focus of evaluation from traditional NLP tasks (Wang et al., 2019b;a) to specific capabilities such as reasoning (Clark et al., 2018; Hendrycks et al., 2021a;b; Rein et al., 2023), coding (Chen et al., 2021; Austin et al., 2021; Cassano et al., 2023; Liu et al., 2023a), multilinguality (Shi et al., 2023), tool use (Srinivasan et al., 2023; Patil et al., 2023; Li et al., 2023; Yan et al., 2024), long context (Shaham et al., 2023; Kamradt, 2023; Zhang et al., 2024; An et al., 2024), image recognition (Yue et al., 2024), instruction following (Zhou et al., 2023), mastering domain-specific knowledge (Hendrycks et al., 2021a), and weakness identification (Chen et al., 2024). Moreover, benchmarks like BiGBench Bench assess a range of abilities across multiple tasks but still target individual capabilities in isolation (Lin et al., 2024; Kim et al., 2024). Although LLMs are capable of handling increasingly complex tasks, cross-capability evaluation remains underexplored. Our work fills this gap by systematically investigating these crucial yet often overlooked cross capabilities.

Another emerging area is the evaluation of LLM-based agents, which inherently require cross capabilities to function effectively in real-world applications. Unlike standalone LLM evaluation, which focuses on specific skills, agent assessment emphasizes the overall success rate in completing tasks (Yao et al., 2022; Zhou et al., 2024; Koh et al., 2024; Liu et al., 2024; Xie et al., 2024) or executing particular actions (Deng et al., 2023; Ma et al., 2024). While CROSSEVAL is designed for LLMs rather than agents, it still encompasses key agent-related capabilities such as multi-modality, multilingualism, and tool use. Furthermore, it provides a comprehensive distinction between individual and cross capabilities, providing a more granular framework for evaluation and analysis.

**Evaluation Metrics for Open-Ended Generation.** Evaluation metrics have evolved alongside advances in model generation capabilities, moving from n-gram-based measures (Papineni et al., 2002; Lin, 2004) to pre-trained language model (PLM)-based evaluators (Zhang et al., 2020; Sellam et al., 2020; Yuan et al., 2021; Zhong et al., 2022) and, more recently, to LLM-as-a-Judge frameworks (Liu et al., 2023b; Zheng et al., 2023). Given the large set of complex, open-ended prompts in our benchmark, we employ LLMs as evaluators to assess model outputs. Unlike previous methods that rely on self-generated prompts, we adopt a point deduction-based prompting technique. Each instance is supported by three expert-annotated references to ensure reliability. Furthermore, CROSSEVAL is also the largest benchmark available for measuring the correlation between LLM ratings and human judgments, providing insights into which capabilities different LLMs excel at evaluating.

## 7 CONCLUSION

We systematically investigated the cross capabilities of LLMs by introducing CROSSEVAL, a testbed designed to evaluate both individual and cross capabilities. We also developed an LLM-based judge that showed strong agreement with human judgments. Our experiments revealed that LLMs consistently follow the "Law of the Weakest Link," where cross-capability performance is limited by the weakest ability, even after enhancing individual abilities. Our benchmark and findings highlight the importance of focusing on cross-capability development and evaluation in future LLM research.

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

**Appendix Table of Contents**

## A   LIMITATIONS

### A.1   REFLECTIONS ON MULTILINGUALISM

Our work contributes to the ongoing scholarly conversation about how to meaningfully evaluate multilingual capabilities in LLMs. In designing our framework, we face important methodological choices regarding the treatment of languages:

- For English capabilities, we follow established conventions from prior work (Llama Team, 2024; Kim et al., 2024), treating English as a foundational capability that encompasses knowledge-based QA and instruction following. This reflects the current reality of language model development, where English often serves as the primary training language and evaluation benchmark.
- The field currently lacks consensus on how to conceptualize non-English languages in evaluation frameworks. Specific approaches (Llama Team, 2024; Kim et al., 2024) aggregate all non-English languages as a single capability, while others, like Chatbot Arena (Zheng et al., 2023), evaluate each language independently.

In our work, we select Spanish as a representative language for studying cross-capability interactions, following the approach used in SEAL LLM Leaderboards[1]. This choice allows us to explore rich interactions between language and other capabilities (e.g., *Spanish & Reasoning*, *Spanish & Image Recognition*).

We acknowledge that this design choice introduces an important limitation: our findings on cross-capability interactions specifically pertain to Spanish and may not generalize to other languages with different linguistic properties, cultural contexts, or representation in training data. Future work expanding this analysis to diverse language families can provide a more comprehensive understanding of how linguistic diversity interacts with other model capabilities. This represents an exciting opportunity to develop more nuanced evaluation frameworks that better reflect the true complexity of multilingual understanding.

### A.2   REFLECTIONS ON MULTIMODALITY

This paper primarily focuses on pure-text scenarios, with 6 out of 7 individual capabilities and 5 out of 7 cross-capabilities being text-only. This focus is reflected in our title and conclusions, which specifically address LLMs rather than vision-language models. We include image recognition as a representative multimodal capability to explore whether the "Law of the Weakest Link" extends beyond text-only interactions, providing an initial reference point as state-of-the-art LLMs increasingly support multimodal inputs.

Our findings regarding multimodal interactions represent preliminary explorations rather than comprehensive evaluations of dedicated vision-language models. As the field continues to advance toward more sophisticated multimodal architectures, future work can expand this analysis to develop more nuanced understandings of cross-capability interactions across diverse modalities.

### A.3   REFLECTIONS ON CAPABILITY RELATIONSHIPS

Our exploration of capability interactions reveals an intriguing aspect of language model behavior that invites deeper consideration: the varying relationships between different capabilities. While we did not explicitly quantify these relationships, our work acknowledges their importance in understanding cross-capability performance:

1) For individual capabilities: Our selection embraces a spectrum of relatedness that enriches our analysis.

- Some capabilities naturally share conceptual foundations (e.g., languages with similar linguistic structures), offering insights into how related skills influence one another.
- Other capabilities represent fundamentally different cognitive domains (e.g., coding versus image recognition), allowing us to observe how models bridge disparate knowledge systems.

---

[1]https://scale.com/leaderboard

2) For cross capabilities: Our choice of cross-capability scenarios is guided by two main criteria:

- **Coverage**: Ensuring all individual capabilities are covered in cross-capability scenarios.
- **Real-world relevance**: Prioritizing combinations that frequently appear in actual user queries and applications.

Perhaps most compelling is how the "Law of the Weakest Link" emerges consistently across this varied landscape of capability relationships. This consistency suggests a fundamental principle of current language models rather than an artifact of particular selections. Future research might develop formal metrics for capability distance, potentially revealing how the strength of this effect varies with the conceptual proximity of different capabilities, a direction that could significantly advance our understanding of emergent behaviors in increasingly capable models.

## A.4 REFLECTIONS ON CAPABILITY COMPARABILITY

In this work, we explore methods to address the challenge of making different capabilities meaningfully comparable, an ongoing question in LLM evaluation that requires careful consideration:

1) **Connecting individual and cross capabilities:** We design our evaluation so cross capabilities incorporate their component individual capabilities, creating natural relationships for comparison. For instance, "Coding & Reasoning" includes coding tasks, establishing connections that help illuminate how capabilities transform when combined.

2) **Consistency in difficulty and style:** We strictly control the difficulty and style distribution of prompts across all capabilities. By standardizing these factors, we ensure that comparisons are valid.

3) **No comparisons of unrelated capabilities:** We do not compare unrelated capabilities with no overlap, as we agree that such comparisons would lack significance. Our evaluations focus on scenarios where meaningful relationships exist, such as cross capabilities that naturally combine individual capabilities used in tandem for real-world tasks.

4) **Consistent patterns:** Across all evaluated scenarios, involving 14 capabilities and 17 models, our results consistently demonstrate the "Law of the Weakest Link." This robust and recurring observation reinforces the significance of our comparisons, revealing systematic trends in model performance and interactions between individual and cross capabilities.

We acknowledge, as reviewers insightfully noted, that establishing perfect comparability between different capabilities requires more in-depth research. Human evaluation could provide valuable evidence for the validity of cross-capability comparisons by demonstrating consistent judgment standards. Future work might incorporate domain experts to provide human labels that validate score comparability across different capabilities, perhaps through calibration studies that establish common scales. These advancements would build upon our exploratory work to develop increasingly robust foundations for understanding how capabilities interact within language models, potentially revealing even more nuanced patterns than we have observed in this initial investigation.

## B    Complete Taxonomy

To ensure the comprehensiveness of the prompt sets in our evaluations, we build taxonomy with Level-1 (L1) and Level-2 (L2) categories. More concretely, Tables 4 – 9 and Tables 10 – 12, present the taxonomy for individual capabilities (*English and Multilingual, Reasoning, Coding, Image Recognition, Tool Use, and Long Context*) and cross capabilities (*Coding & Reasoning*, *Image Recognition & Reasoning*, *Tool Use & Coding*, *Tool Use & Reasoning*, *Long Context & Coding*), respectively.

### B.1    Taxonomy of Individual Capabilities

| L1 Categories | L2 Categories |
|---|---|
| Factual Questions about Recent and Current Things | Historical events & figures
Scientific concepts and explanations
Geographical information
Cultural & social topics
Technical information |
| Very Accurate Questions (Beyond Expected Model Knowledge) | Historical events & figures
Scientific concepts and explanations
Geographical information
Cultural & social topics
Technical information |
| Procedural Questions about Recent, Current, or Local Things | Cooking & food preparation
Home & DIY projects
Technology & devices
Arts & crafts
Travel & transportation
Work & productivity
Health & fitness |
| Recommendations / Brainstorming about Local and Current Things | Dining & food suggestions
Entertainment suggestions
Travel & destinations suggestions
Product & service recommendations |
| Tasks with File Uploads | Content Summarization
Question Answering |

Table 4: **Taxonomy of the tool use capability.**

| L1 Categories | L2 Categories |
|---|---|
| Factual Questions | Historical events & figures
Scientific concepts and explanations
Geographical information
Cultural & social topics
Technical information |
| Procedural Questions | Cooking & food preparation
Home & DIY projects
Technology & devices
Arts & crafts
Travel & transportation
Finance & budgeting
Work & productivity
Health & fitness |
| Language Assistance | Grammar, spelling, & vocabulary |
| Writing & Content Creation | Analysis
Creative writing: Fiction
Creative writing: Poetry and Songwriting
Creative writing: Social media posts
Creative writing: Nonfiction
Business writing
Legal writing
Classification
Summarization & editing |
| Dialogue | Identity / Personas
Chit-Chat
Advice
Games: Choose-your-own-adventure
Games: Word & language
Games: Social & party |
| Recommendations / Brainstorming | Dining & food suggestions
Entertainment suggestions
Travel & destinations suggestions
Product & service recommendations |
| Personal Growth and Development | Build confidence and self-esteem
Emotional support
Goal setting
Motivation
Physical health support
Professional and career support
Relationship support
Tutoring and learning support |
| Social Interaction and Communication | Debate and opinions
Discuss shared interests
Humor and jokes
Socialize with friends (group chat) |

Table 5: **Taxonomy of the English and multilingual capabilities.**

| L1 Categories | L2 Categories |
|---|---|
| Mathematical Calculation | Arithmetic & basic math
Algebra & equations
Geometry & trigonometry
Calculus & advanced math
Probability & statistics
Discrete math & logic
Ordinary and partial differential equations |
| Mathematical Reasoning | Math word problem solving
Math question answering
Theorem proving (e.g. proofs)
Mathematical model building |
| Commonsense Reasoning | Physical reasoning
Temporal reasoning
Spatial reasoning |
| Logic / Problem Solving | Identifying root causes & issues
Evaluating evidence & reasoning
Identifying pros & cons
Inductive reasoning
Deductive reasoning |
| Social and Emotional Reasoning | Empathy and perspective taking
Social norm understanding
Humor understanding
Negotiation
Emotion recognition / sentiment analysis |
| Moral and Ethical Reasoning | Consequence evaluation
Applying moral and ethical principles
Resolving moral or ethical dilemmas (conflict of principles) |
| Scientific Reasoning | Hypothesis formation and testing
Causal reasoning
Scientific evidence evaluation
Model-based reasoning |
| Legal Reasoning | Case-Based Reasoning
Statutory Interpretation
Contract Interpretation
Administrative Regulation Interpretation
Legal Evidence Evaluation |

Table 6: **Taxonomy of the reasoning capability.**

| L1 Categories | L2 Categories |
|---|---|
| Code Generation / Synthesis | Code generation (Text to Code)
Code completion
Code Summarization / Compression
Code to Code (same language)
CLI
Coding Ecosystem
Code to Code (different languages) |
| Code Documentation | Comment generation
Commit text generation
Document this function
Create example usages of this function
Create API documentation |
| Code Debugging | Debugging & troubleshooting
Testing |
| Code Review & Best Practices | Code review
Security Review
Quality Assurance
Log Analysis (Text to Text) |

Table 7: **Taxonomy of the coding capability.**

| L1 Categories | L2 Categories |
|---|---|
| Object Recognition | Single Object Recognition
Multiple Object Recognition
Fine-Grained Object Recognition |
| Scene Understanding | Indoor Scene Understanding
Outdoor Scene Understanding
Cultural Scene Understanding
Complex Scene Understanding |
| Image Captioning | Descriptive Captioning
Abstract Captioning |
| Attribute and Relationship Identification | Object Attribute Identification
Spatial Relationship Identification
Semantic Relationship Identification |
| Dialogue | Visual How to
Memes |
| Graceful Refusals | Vague or unrelated question
Blurry image
Unsupported capabilities |

Table 8: **Taxonomy of the image recognition capability.**

| L1 Categories | L2 Categories |
|---|---|
| Factoid or Complex Question Answering | Scientific Documents
Financial Documents
Books
Legal Documents
Podcast transcripts
Video/Movie transcripts |
| Summarization | Scientific Documents
Financial Documents
Books
Legal Documents
Podcast transcripts
Video/Movie transcripts |
| Multi-Document Understanding (Q&A) | Home & personal
Work & business |

Table 9: **Taxonomy of the long context capability.**

## B.2 TAXONOMY OF CROSS CAPABILITIES

| L1 Categories | L2 Categories |
|---|---|
| Coding Q&A (Text to Text) | Programming concepts & guidance
Software Architecture
Language-specific features
Code summarization
Frameworks & tools |
| Code Explanation | Code walkthroughs
Algorithm explanations |
| Programming Assistant | Code Understanding
Problem decomposition
Algorithmic reasoning
Debugging reasoning
Code optimization |
| Mathematical Calculation | Arithmetic & basic math
Algebra & equations
Geometry & trigonometry
Calculus & advanced math
Probability & statistics
Discrete math & logic
Ordinary and partial differential equations |

Table 10: **Taxonomy of the coding & reasoning capability.**

| L1 Categories | L2 Categories |
|---|---|
| Diagram Understanding | Scientific Diagram Understanding
Flowchart Understanding
Graph Understanding |
| Chart Understanding | Basic Chart Understanding (Localization)
Basic Chart Descriptions
Chart reasoning |
| Text-Rich Understanding | Document understanding
Others |
| Visual Math and Science | Formula understanding
Figure understanding |

Table 11: **Taxonomy of the image recognition & reasoning capability.**

| L1 Categories | L2 Categories |
|---|---|
| Repository-Level Code Generation | Code generation (Text to Code)
Code completion
Code Summarization
Code to Code (different languages)
Code modification |
| Repository-Level Code Understanding | Code Q&A / summarization
Code walkthroughs |
| Repository-Level Code Debugging | Debugging & troubleshooting |
| Log Analysis | Parsing logs into structured templates
Finding anomalies from raw logs
Detecting errors and debugging suggestions |
| API Docs Understanding | Q&A on API
Code generation with API |

Table 12: **Taxonomy of the long context & coding capability.**

| L1 Categories | L2 Categories |
|---|---|
| Code Execution | Code generation and execution (Text to Code)
Code to Code (Same language)
Create example usages of this function |
| Code Debugging with Execution | Debugging, troubleshooting, and optimizing code
Testing |
| Programming Assistant with Execution | Code Understanding |
| Code Execution with File Uploads | Data Analysis
Data Visualization
Code Review / Explanation / Debugging |

Table 13: **Taxonomy of the tool use & coding capability.**

| L1 Categories | L2 Categories |
|---|---|
| Mathematical Reasoning | Math word problem solving
Math question answering |
| Scientific Reasoning | Physics
Chemistry
Units and Measures
Computational Sciences
Earth Sciences
Materials
Space and Astronomy
Life Sciences
Technological World
Weather and Meteorology
Food Science
Transportation
Health and Medicine
Physical Geography
Engineering |
| Mathematical Calculation | Arithmetic & basic math
Algebra & equations
Geometry & trigonometry
Calculus & advanced math
Probability & statistics
Discrete math & logic
Number Theory
Linear Algebra
Plotting
Complex Analysis
Continued Fractions
Trigonometry
Ordinary and partial differential equations |

Table 14: **Taxonomy of the tool use & reasoning capability.**

### B.3  DETAILS FOR TAXONOMY CONSTRUCTION

The manual taxonomy construction follows a multi-stage annotation process, detailed as follows:

**Annotation Team.**  The annotation team consists of contributors with experience as LLM researchers or product managers, including some of the authors of this paper. The team's diverse expertise ensures that the taxonomy captures a wide range of real-world use cases.

**L1 Annotation.** Each capability's L1 categories are derived from a large set of real user queries:

- Three team members independently annotate the L1 categories by identifying common tasks in the queries.
- After the independent annotations, the team convenes to merge similar categories, address any omissions, and finalize the L1 taxonomy collaboratively.

**L2 Annotation.** Building on the agreed-upon L1 categories, the annotators independently propose L2 subcategories:

- These finer-grained tasks are designed to provide additional specificity while maintaining consistency with the L1 framework.
- A second round of discussions and revisions is conducted to refine the L2 categories and reach a consensus.

**Team Review.** Once the L1 and L2 categories are finalized, all taxonomies undergo a comprehensive review by the full team:

- The team fine-tunes the taxonomy to ensure that all breakdowns adhere to a consistent granularity across capabilities and align with the MECE principle (mutually exclusive and collectively exhaustive).

**Annotation Effort.** The annotation effort involves more than ten main contributors and spans approximately one week.

# C MORE DETAILS ON CAPABILITIES

## C.1 PRINCIPLES FOR DEFINING CAPABILITIES

In our framework, a capability is defined by the following principles:

- **Distinctiveness:** The capability should have unique characteristics that distinguish it from others.
- **Measurability:** The capability should be independently evaluatable through specific tasks.
- **Consistency:** The capability should maintain its fundamental characteristics across different contexts and domains.

Using these principles, we define the seven individual capabilities as described in Section §2.1.

## C.2 DISTINCTION BETWEEN CAPABILITY AND SKILL

The concept of "skill" is relatively vague and can manifest in different ways:

- It can be a subset of a capability, fitting within our taxonomy (e.g., "basic math formulas" and "analytical reasoning" as part of Reasoning capability).
- It may span multiple capabilities. For instance, "research" can be a skill that may require various capabilities: *Tool Use* (to gather information via web browsing), *Long Context* (to process and synthesize extensive documents), *Reasoning* (to evaluate and draw conclusions from the findings), and potentially *Image Recognition* (to interpret graphs and figures in academic papers).

Given this variability, we follow the capability-based evaluation approach commonly used in previous studies and utilize our manually created taxonomy to distinguish the boundaries of different capabilities.

## D  CROSSEVAL BENCHMARK

### D.1  WHY CONSTRUCTING THE BENCHMARK FROM SCRATCH

While existing benchmarks for specific core capabilities exist, we find it necessary to construct prompts for the following reasons:

1) **Lack of Clear Categorization:** Existing benchmarks often lack precise categorization for individual prompts. For example, benchmarks evaluating coding tasks may include prompts that, according to our taxonomy, fall under either the individual capability "Coding" or the cross-capability "Coding" & "Reasoning." Without this categorization, it is impossible to conduct the in-depth analyses needed to understand the relationships between individual and cross capabilities.

2) **Coverage:** Existing benchmarks do not cover all 76 Level-1 and 332 Level-2 categories in our taxonomy. To ensure thorough evaluation, we construct prompts to fill these gaps and provide more complete coverage.

3) **Difficulty Distribution:** Existing benchmarks often lack consistent difficulty distribution or use varying distributions across tasks. These inconsistencies hinder systematic comparisons of individual and cross capabilities. Our benchmark standardizes difficulty levels across all capabilities, supporting meaningful analysis.

4) **Style Distribution:** Existing benchmarks vary significantly in prompt styles, such as QA formats, exam-like questions, or real-world scenarios. These variations hinder consistency in evaluation. To address this, we standardize all prompts in our benchmark to reflect open-ended, real-world user queries, ensuring uniformity across capabilities.

### D.2  SELECTION PROCESS AND CRITERIA FOR PROMPTS

**Selection of 100 prompts.** Our reviewers follow a strict quality screening process to ensure all prompts align with the task categories in our taxonomy and meet the criteria outlined in the guidelines. If issues such as non-compliance, incomplete task coverage, or misalignment with the difficulty distribution are identified by our reviewers, the data vendor reannotates the problematic parts. After review and reannotation, the number of annotated prompts for each capability exceeds 100, with a unified quality control standard ensuring high-quality prompts across all capabilities.

**Why limit to 100 prompts.** We limit 100 prompts for each capability for the following reasons:

- Consistency: Standardizing to 100 prompts per capability ensures fair comparisons, preventing any capability from being over- or underrepresented due to variations in dataset size.
- Resource Efficiency: Each prompt requires multiple model responses and expert annotations, which are resource-intensive. Limiting to 100 prompts maintains quality while managing costs and reduces the overall expense of evaluation on our benchmark.

### D.3  STATISTICS OF THE PROMPT SET

The prompt set annotated in CrossEval covers all 76 Level-1 and 332 Level-2 categories listed in the previously mentioned taxonomy, with specific statistics provided in Table 15.

| | Capabilities | # Prompts | # L1 Categories | # L2 Categories |
|---|---|---|---|---|
| Individual | English | 100 | 8 | 45 |
| | Reasoning | 100 | 8 | 36 |
| | Coding | 100 | 4 | 18 |
| | Image Recognition | 100 | 6 | 17 |
| | Tool Use | 100 | 5 | 23 |
| | Long Context | 100 | 3 | 14 |
| | Spanish | 100 | 8 | 45 |
| Cross | Coding & Reasoning | 100 | 4 | 19 |
| | Image Recognition & Reasoning | 100 | 4 | 10 |
| | Tool Use & Coding | 100 | 4 | 9 |
| | Tool Use & Reasoning | 100 | 3 | 30 |
| | Long Context & Coding | 100 | 5 | 13 |
| | Spanish & Reasoning | 100 | 8 | 36 |
| | Spanish & Image Recognition | 100 | 6 | 17 |

Table 15: **Statistics of the prompt sets in the CROSSEVAL benchmark.**

## D.4   PROMPT SET EXAMPLES

To provide an intuitive sense of the types and difficulty of the prompt set in our benchmark CROSS-SEVAL, we present examples for each capability, including the difficulty level, L1 and L2 categories, and the prompts. Tables 16 – 22 correspond to individual capabilities, while Tables 23 – 27 pertain to cross capabilities.

| Difficulty | L1 Category | L2 Category | Prompt |
|---|---|---|---|
| Easy | Logic / problem solving | Deductive reasoning | All bachelors have never married. John is a man who has never been married. Is John a bachelor? |
| Easy | Commonsense Reasoning | Spatial reasoning | If you enter a building from the east side, walking west, and then take two rights and a left down corridors, which direction are you facing? |
| Medium | Mathematical Calculation | Discrete math & logic | Jane won the lottery and decided to spend some of the money. She spent $1.50 on the first day. She spent $3 on the second day. She spent $4.50 on the third day. She kept spending her winnings in the same pattern and then on the last day, she spent her remaining $300. How much did she win in the lottery? |
| Medium | Social and Emotional Reasoning | Empathy and perspective taking | I have a member on my team that is not pulling his weight and I am thinking about firing him. I heard from another colleague that he may be going through a divorce, but he should not allow this to affect his work. Our team is taking a huge productivity hit. What should I do? Explain it to me. |
| Hard | Mathematical Calculation | Ordinary and partial differential equations | Solve the initial value problem: $\frac{1}{2}u_{xx} - u_y = \frac{2}{x^2}, \quad u(x,0) = x.$ You will find that the solution blows up in finite time. Explain this in terms of the characteristics for this equation and explain your reasoning step by step. |
| Hard | Social and Emotional Reasoning | Humor understanding | Two chemists are sitting at a bar. The first chemist tells the bartender, "I'll have some H2O." The second chemist tells the bartender, "I will also have some water". The first chemist tells the second chemist, "darn my murder plot failed". Please explain this joke to me |
| Hard | Mathematical Reasoning | Mathematical model building | One interesting and complex problem that can be addressed through mathematical modeling in fisheries biology is the effectiveness of fish stocking to increase angling opportunities.

Problem Statement: Optimize the amount of angling opportunities by introducing the correct number of bass fish to a given lake.

Variables to Consider:
x: Size of given body of water.
v: volume of vegetation growing in body of water.
y: amount of forage fish per acre via sampling data.
c: number of hours in angling pressure on given body of water per month. |

Table 16: **Examples of the prompt set for reasoning capability.**

| Difficulty | L1 Category | L2 Category | Prompt |
|---|---|---|---|
| Easy | Procedural Questions | Technology & devices | I'm having trouble setting up my router. I want to change the stock password for our home network, and also create a guest network with a separate password for my friends and family. Can you help me change my password, and create a separate guest network with a new password? |
| Easy | Recommendations / Brainstorming | Product & service recommendations | My wife is really into her arts and crafts. She loves painting in her spare time. Can you please recommend something I could get her as a present? Tell me 6 candidates. |
| Medium | Dialogue | Identity / Personas | You are Christopher Walken. I am Sylvester Stallone. Create a series of sentence openers about our movies that I can respond to. try and make them serious so I can try and make you laugh. |
| Medium | Writing & content creation | Analysis | In Noo Sara-Wiwa's book about traveling to Nigeria "Transwonderland", how does her outlook on Nigeria change as her journey through the country progresses? Include descriptions of her first impressions when traveling from one region/city of Nigeria to another and describe her feelings as someone who has Nigerian heritage but moved to England at a very young age. Use vivid language in your response. |
| Hard | Factual Questions | Technical information | My question relates to 3d printing. I'd like to understand more about the chemical differences between two printing materials – PLA and TPU. Start by explaining the chemical differences. Then talk about the physical properties of both materials, listing three to five use cases for each. Finally, give me an insight into practical considerations when printing with these materials. I'm particularly interested in recommended extruder and printing bed settings.
- I'm looking for a detailed response written in clear and simple English.
- I'm a novice, so be sure to italicize any technical terms and provide a definition in parentheses.
- Please keep your response to 600 words, give or take 10 percent.
- Separate sections and sub-sections with H1 and H2 headers. |
| Hard | Dialogue | Games: Choose-your-own-adventure | You are a game developer, focused on choose-your-own-adventure style text-based games. You are creating a main character for the story and need to finalize aspects of the character's personality. The main character is a young male, roughly 14 years of age. He is a wizard with science and technology subjects, and his personality should reflect this. He has so far chosen routes that lead the character down a positive route, with a general increase in his knowledge and skills. He is unlucky in love but romance is one of the major focuses of the narrative. What sort of creative personality would the character have that allows the player to connect with and better feel themselves in the role as they choose the paths they are going to take? List five less common ones. Also, please provide a variety of routes in this game that might change this personality. Make sure you provide at least 3 routes with each one being around 150 words. At least one route should have negative consequences for the character's personality, and one should be decisively positive. Provide a name for this character that reflects the personality you choose for them. |
| Hard | Social interaction and communication | Humor and jokes | I need to write a short stand-up comedy routine for a friend's dinner party. It should take no longer than 4 minutes to perform. The audience will consist of a baker, a doctor, and a florist, so try and make jokes relevant to them. I can do a good impression of Homer Simpson, so please write it in his style. The tone should be silly and playful, but be sure not to make fun of the audience. |

Table 17: **Examples of the prompt set for English capability.**

| Difficulty | L1 Category | L2 Category | Prompt |
|---|---|---|---|
| Easy | Code review & Best practices | Quality Assurance | Find any issues with the following code and suggests any potential fixes. Our goal with the code is to keep drawing lottery tickets (where it is a lottery game where you match 6 numbers out of 49 numbers) until there is a winning ticket. We want to print out the time it takes and the number of draws until we draw the winning ticket, and then print the actual winning ticket along with the theoretical expected number of draws until success.

*{attached code}* |
| Easy | Code generation / synthesis | Coding Ecosystem | What command is used to add a package to a rust project? How do you fix "error[E0433]: failed to resolve: could not find 'quote' in the crate root" or "error[E0432]: unresolved import 'quote'"

*{attached code}* |
| Medium | Code documentation | Create API documentation | Create an API documentation for this Python code that utilizes Flask for API routes.

*{attached code}* |
| Medium | Code debugging | Testing | I want to update the data I have determined in the database by pulling it with post. However, it is not updated. The codes in the view.py file are also added. It may work for the application. Also ensure to write unit tests for the function.

*{attached code}* |
| Hard | Code documentation | Comment generation | Create a readme document that breaks down what the code does, and the the various techniques that it uses. Also printout some example usage of the program, displaying the result/solution to the example data in code. Also include a section breaking down the time/space complexity of the program, and how the efficiency of this program compares to other methods without any optimizations/techniques. The document should read at an undergrad level.

*{attached code}* |
| Hard | Code generation / synthesis | Coding Ecosystem | Please create a React custom hook that can store, retrieve, and sync data from the browser's local storage to the React component that uses the hook. Use the localStorage API. Hook should be called useLocalStorage. When the component mounts, it should read the local storage value as the initial value, and update local storage when the component's state changes, and listen for changes in local storage and update the component state accordingly. The hook's API should look like useState. |
| Hard | Code generation / synthesis | Code completion | I've been working on implementing some computer vision algorithms. I've written the Harris corner detection for finding good features. I want some other feature finding algorithms: FAST and ORB features. Can you finish those functions? Also, I want to implement Lucas-Kanade optical flow. I have a rough idea of what the function will take in, can you also finish this for me?

*{attached code}* |

Table 18: **Examples of the prompt set for coding capability.**

| Difficulty | L1 Category | L2 Category | Image | Prompt |
|---|---|---|---|---|
| Easy | Image Captioning | Descriptive Captioning |  | Describe every creature in this image. |
| Easy | Graceful refusals | Blurry image |  | What does the license plate say? |
| Medium | Attribute and Relationship Identification | Object Attribute Identification |  | What is the color and shape of the "10" ball in the photo? |
| Medium | Object Recognition | Multiple Object Recognition |  | How many of these dogs have floppy ears? How many of the floppy-eared dogs have black fur? |
| Hard | Comprehensive use case | Memes |  | Explain this meme in detail. |
| Hard | Scene Understanding | Indoor Scene Understanding |  | What type of person would you imagine lives in this room? Pick 5 items to justify your answer. |
| Hard | Comprehensive use case | Visual How to |  | My bike is not ridable. What does it need to be fixed and how do I fix it? |

Table 19: **Examples of the prompt set for image recognition capability.**

| Difficulty | L1 Category | L2 Category | Prompt |
|---|---|---|---|
| Easy | Recommendations / Brainstorming about Local and Current Things | Entertainment suggestions | What are some fun things I can watch on local channels in Henry County, Illinois today? |
| Easy | Tasks with File Uploads | Content Summarization | Summarize the below content in the attached file in 150 words or less, focusing on the mode of action, possible adverse effect's and effectiveness. 

 {*attached file*} |
| Medium | Procedural Questions about Recent, Current, or Local Things | Home & DIY projects | What are the step by step instructions of installing a Black And Decker BD05MWT6 Window Air Conditioner, 5000 BTUs, unit? Will it fit into a window with the dimensions of 24 inches wide and 48 inches height? |
| Medium | Factual Questions about Recent and Current Things | Cultural & social topics | Show me the current top 10 hockey players in the USA, what the game statistics are for each person, and how many awards they have won. |
| Hard | Very accurate questions (beyond expected model knowledge) | Scientific concepts and explanations | Please list all recent coronal mass ejections from the past 365 days and what their magnitude was. Were there any correlating changes in the schumann resonance? Does science recognize effects of coronal mass ejections from the sun on the schumann resonance? |
| Hard | Tasks with File Uploads | Question Answering | The text mentions several individuals and their contributions or reputations in 1817. Choose three and discuss how they are represented and their significance in the broader historical or cultural context of that year. Also, what role does satire and public opinion play in the text, particularly in the portrayal of political figures and events? Provide specific examples from the given file. 

 {*attached file*} |
| Hard | Procedural Questions about Recent, Current, or Local Things | Arts & crafts | I want to re-cover a pair of wingback chairs. I love the idea of painting or printing my own fabric. Can you give me a list of supplies for this? I also need a step-by-step walkthrough of easy ways to paint or transfer scenes to fabric.-a selection of two or three comprehensive youtube videos would be really helpful. I'm also open to other ideas like stamping or stenciling. GIve me a list of methods that won't break the bank. Finally, I need step-by-step instructions for re-covering them as well. Please list any fabric and/or craft stores near me, their hours of operation. List them in the order of most to least likely to have the supplies I need. |
| Hard | Factual Questions about Recent and Current Things | Geographical information | Make a table of all countries in which at least 25% of the current population speaks French. The table should include the name of the country, their current population, the percentage of that population who speak French, and the name of the current head of state. Below the table, summarize trends in the percentage of French language speakers globally over the past four years. This summary should be no more than 150 words. Finally, list any examples of French-language radio stations, French-language newspapers, and French-language television stations currently operating in countries in which French is not the predominant national language. Provide links websites for those entities wherever possible. |

Table 20: **Examples of the prompt set for tool use capability.**

| Difficulty | L1 Category | L2 Category | Prompt |
|---|---|---|---|
| Easy | Factoid or Complex Question Answering | Podcast transcripts | Given this podcast transcript, please summarize Daniel Dennett's opinion on Gould's idea of Non-Overlapping Magisteria, and explain how Dennett believes science ought to fit into society. Give a specific quote to support your answer to each question. 

 {*attached text*} |
| Easy | Multi-Document Understanding | Home & personal | What are some of the most common pieces of advice for maintaining a vegetable garden. Is there any information that is controversial? 

 {*attached text*} |
| Medium | Summarization | Financial Documents | Based on the financial documents of JB-HIFI in 2021, summarise the Notes to the financial statements and note the financial statement where each note relates to. 

 {*attached text*} |
| Medium | Factoid or Complex Question Answering | Books | What caused Jane to be locked up in the "red-room"? How did the "red-room" affect her adult life in both positive and negative ways? Did she experience any flashbacks regarding the "red-room" in her adult life? 

 {*attached text*} |
| Hard | Summarization | Books | Summarize every four chapters of this book in three paragraphs. The first paragraph should summarize the first and second chapters, the second paragraph should summarize the third and fourth chapters, and the third paragraph should focus on the fifth chapter. Create a title for each section in 1-3 words that encapsulates the events of these chapters. Repeat this structure for each subsequent set of five chapters! 

 {*attached text*} |
| Hard | Factoid or Complex Question Answering | Legal Documents | Given the segment of the NBA CBA which outlines financial rules, how might strategy differ in terms of team/roster construction with salaries for a contending team differ from a lottery team? Justify you response with quotes from the text, and give examples with a fictional roster with salaries to demonstrate the salary repercussions/strategy being proposed. 

 {*attached text*} |
| Hard | Multi-Document Understanding | Work & business | Given the latest annual financial reports from AMD (Advanced Microdevices) and NVIDIA, evaluate each company's strategy and preparedness for addressing AI workloads over the next 5 years include their comparative headwinds and tailwinds. Assess which company is currently best positioned to gain the most market share in AI and why. 

 {*attached text*} |
| Hard | Factoid or Complex Question Answering | Books | I read this book, but I forgot to take notes. So I understand the philosophy, but I don't remember the exact steps that I should follow. Extract all the exact information that the successful man shared with his two new friends, so I can print these routines, rules, and truths and have them in my sight. I am talking about the 5-3-1 facts or the 20-20-20 routine, etc. 

 {*attached text*} |

Table 21: **Examples of the prompt set for long context capability.**

| Difficulty | L1 Category | L2 Category | Prompt |
|---|---|---|---|
| Easy | Dialogue | Identity / Personas | Digamos que ahorita eres Tesla y me tienes que responder como si fueras el.¿ que me dirias si te pregunto acerca de tus mejores inventos? |
| Easy | Factual Questions | Cultural & social topics | Escribe una síntesis acerca de las características principales del culto mariano en América Latina, con énfasis especial en los países más grandes y religiosos: México y Brasil. |
| Medium | Writing & content creation | Creative writing: Fiction | Escribe una historia de animales similar a la película de Madagascar pero ambientada en la cueva Hang Son Doong de Vietnam. Los tres personajes principales deben ser un murciélago, un pájaro y un escorpión. Las estalactitas y estalagmitas deben poder hablar. El villano de la historia debe ser un gusano bioluminiscente. |
| Medium | Procedural Questions | Technology & devices | Estoy teniendo problemas con mi laptop. Es una Dell Latitude 3440 con Windows 10. Cada tanto tiempo, entre 5 minutos y una hora, se desconecta el Wi-Fi y debo volverlo a conectar manualmente. Dame 5 posibles soluciones, en viñetas. Las soluciones deben ser totalmente detalladas, como para alguien que no tiene conocimiento de computación. |
| Hard | Social interaction and communication | Discuss shared interests | Me encanta leer, especialmente libros de ficción. Sin embargo, este año quiero empezar a leer otros tipos de libros. Me puedes compartir una lista de tus 15 libros favoritos que no sean de ficción? enfócate en autores latinoamericanos o asiáticos y estoy abierta a cualquier tipo de escritura (poemas, biografías, etc.), dime el idioma original y la nacionalidad de cada autor y si tiene traducción al español en caso de que sea necesario |
| Hard | Procedural Questions | Finance & budgeting | Necesito ahorrar más en los gastos misceláneos y de entretenimiento digital. Establece un presupuesto mensual para cocinar la mayoría de las comidas en casa. Realiza las recomendaciones teniendo en mente a una persona que vive en el suroeste de Florida, de manera que los mercados donde conseguir comida saludable y barata estén basados en la disponibilidad de dicha región. El gasto máximo mensual será de $400, y debes especificar como distribuirlo. En cuanto a los gastos de entretenimiento digital, he identificado que la mayoría de ellos están vinculados a suscripciones a plataformas de streaming. Necesito quedarme solo con dos de estas plataformas. Mis intereses principales son los deportes y el cine clásico. Recomiéndame las que mejor puedan satisfacer esta demanda dentro de un presupuesto limitado. Lo máximo que puedo gastar en entretenimiento digital son $21. |
| Hard | Recommendations / Brainstorming | Product & service recommendations | Soy una aficionada del trekking y de acampar en la naturaleza. Lamentablemente, a veces me resulta difícil salir a caminar, sobre todo a lugares fríos porque no cuento con suficiente equipo de calidad. La ropa de trekking especializada se ha vuelto muy cara, así como las botas de trekking, las bolsas de dormir, tiendas de campaña, etc etc. Por favor, ayúdame con recomendaciones para un equipo básico, que sea de buena calidad, pero con precios accesibles. Necesito 3 atuendos completos de trekking, para temperaturas de 0 a 15 grados centígrados. Para la zapatillas, estoy dispuesta a gastar más en un par que me dure al menos 3 años. Qué sean impermeables, cómodas y calientitas. Finalmente, necesito invertir en una bolsa de dormir. ¡Pero una que sea ligera! |
| Hard | Social interaction and communication | Humor and jokes | dime un chiste con más de 300 palabras que incluya las palabras gato, música, trueno y árbol , que sea del punto de perspectiva del gato e incluya una frase final chusca y con moraleja. |

Table 22: **Examples of the prompt set for Spanish capability.**

| Difficulty | L1 Category | L2 Category | Prompt |
|---|---|---|---|
| Easy | Programming Assistant | Code Understanding | I have inherited this python function from a previous Data Scientist in my team who has left. I need to understand what the function is doing - there are no comments and I don't have time to debug this function.

*{attached code}* |
| Easy | Coding Q&A (Text to Text) | Software Architecture | I'm planning to implement different shape classes in Java. They have similar fields, but the parameters are different. For example, a circle has a radius field, but a square has a length field. I want to apply the factory design patterns to my program; is it a good practice? If not, which design patterns suit my requirements more? |
| Medium | Programming Assistant | Algorithmic reasoning | Design the most efficient algorithm to find all unique pairs of integers in a given list that sum to a target value. Ensure that each pair is unique (no pair should repeat even if the integers appear multiple times in the list). Describe the algorithm's logic and implementation details. Explain why you chose to implement it in the way you did in order to achieve the best time complexity and memory storage possible. |
| Medium | Mathematical Calculation | Ordinary and partial differential equations | Suppose that $\frac{dy}{dt} = y + 1$, and $y(0) = 1$. Write me Python code which uses Euler's method to estimate y(1). |
| Hard | Programming Assistant | Problem decomposition | I'm working on a simple command line game, and I want to break it up into some reusable functions, but I'm not sure how to refactor it. Can you suggest any functions that would make my code less repetitive? Also, help me write 5 more useful functions in the code.

*{attached code}* |
| Hard | Coding Q&A (Text to Text) | Programming concepts & guidance | Can you explain "tail recursion" and why it's key for boosting efficiency in recursive functions? It's known for using less memory than standard recursion. Could you demonstrate this with a Python example, showing both a tail-recursive function and a regular recursive function? Also, since Python doesn't automatically optimize tail recursion, how does this limitation impact its use in larger projects? BTW, how do other programming languages handle tail recursion differently, and what advantages they offer? |
| Hard | Mathematical Calculation | Algebra & equations | There are 8 people in a room, and two boxes. One box has 8 hats, 2 pink, 1 red, 3 blue, and 2 black. The other box has 12 pairs of gloves, 5 green, 4 black, 2 orange, and 1 yellow. The 8 people in the room are invited to each grab and put on one hat and one pair of gloves. Then, each person is will shake hands with 5 different people at random.

Create an algebraic equation to find the probability of someone wearing a pink hat and orange gloves shaking hands with someone wearing a black hat and green gloves at least once?

Use a python script to solve the equation to find the probability, given in percentage. |
| Hard | Code Explanation | Code walkthroughs | Can you please explain what the following code does and how the output is represented? Do we really need to represent the output this way? Also, provide me with a way to implement the following algorithm based on other libraries.

*{attached code}* |

Table 23: **Examples of the prompt set for coding & reasoning capability.**

| Difficulty | L1 Category | L2 Category | Image | Prompt |
|---|---|---|---|---|
| Easy | Chart Understanding | Basic Chart Understanding (Localization) |  | From this chart, name at least one manufacturer more popular than Toyota |
| Easy | Visual Math and Science | Figure understanding |  | What is this illusion called? |
| Medium | Chart Understanding | Basic Chart Understanding (Localization) |  | What is the max trading volume in the bar chart? |
| Medium | Visual Math and Science | Formula understanding |  | What is the chemical formula that comes after the word "Methanol"? Provide the text for the formula, and what the formula is representing/used for. |
| Hard | Diagram Understanding | Flowchart Understanding |  | Based on the diagram, how many steps need to do before Xsemble Design? |
| Hard | Text-rich Understanding | Others |  | So what is the best strategy for the interview question in the image? Explain it step by step. |
| Hard | Diagram Understanding | Graph Understanding |  | How many nodes do I have to remove from this graph so that it is possible to visit every vertex without repeating an edge, starting from the one at the bottom-left. |

Table 24: **Examples of the prompt set for image recognition & reasoning capability.**

| Difficulty | L1 Category | L2 Category | Prompt |
|---|---|---|---|
| Easy | Repository-Level Code Generation | Code completion | In the 'RingIntercom class', there is functionality to unlock a door and process notifications. Extend this class by adding a method that logs all unlock actions with a timestamp and the user ID who performed the unlock. This method should save the log to a local file named 'unlock_log.txt'.

*{attached code}* |
| Easy | Log Analysis | Parsing logs into structured templates | Can you parse logs from this log file I provided into a tree-like structure?

*{attached log}* |
| Medium | API Docs Understanding | Code generation with API | Referencing the Notify.lk API documentation, can you develop a simple command-line application in Java that checks the balance of your Notify.lk account and prints it out? Assume you have the necessary API credentials.

*{attached code}* |
| Medium | Repository-Level Code Generation | Code modification | Optimize the application's performance by implementing lazy loading for the React components associated with routes in 'App.js'. Use 'React.lazy' and 'Suspense' from React to lazily load the components for '/profile/:id', '/editor', and '/player/:id'. Also, provide a fallback loading component that displays a loading spinner while the components are being loaded. Provide the code modifications in 'App.js' to implement lazy loading with a fallback.

*{attached code}* |
| Hard | API Docs Understanding | Q&A on API | I'm working on an automatic grader using Python for my intro to OOP class. Currently I still need to download all my student's .py submissions manually and then upload/enter each grade manually. Do you think it would be possible to use the Canvas LMS API to automate these processes? How do I download the submissions for a given student or set/update the grade on an assignment for a student using python? here is some documentation for the canvas API

*{attached code}* |
| Hard | Repository-Level Code Debugging | Debugging & troubleshooting | It seems there is a bug in the code that ensures that the start point for the ray casting algorithm is outside the polygon. This causes the 3D items not being able to be moved after I update the layout (floorplan) of the room. Locate the function that needs to be fixed and suggest a fix.

*{attached code}* |
| Hard | Repository-Level Code Generation | Code generation (Text to Code) | Given repository, create a system to display the top eight high scores for the game session. When the game starts up, the high scores should be [80, 70, 60, 50, 40, 30, 20, 10] all with a name label 'Anonymous'... when the user dies, if their final score beats any on the high scores list, they should be able enter their name and add their highscore to the list in the appropriate place (i.e. all lower scores should be moved down the list and the lowest removed altogether). When the game is not in AppState 'Game', The state should alternate between the intro screen and a highscores screen that displays the high scores list. The high scores screen should consist of text containing the list of high scores displayed over the same background used for the Game and IntroScreen states, text font should be consistent with the other text in the game and should be white unless it is a new highscore, in which case it that entry should be green.

*{attached code}* |

Table 25: **Examples of the prompt set for long context & coding capability.**

| Difficulty | L1 Category | L2 Category | Prompt |
|---|---|---|---|
| Easy | Code Execution with File Uploads | Data Visualization | Group this sales data by month using pandas and then create a bar chart of the sales per month. Run the code and save the results.

*{attached file}* |
| Easy | Code debugging with execution | Testing | Can you generate a set of unit tests for each method? Please have 10 different and valid test cases for each method in the given code. Make sure to cover edge cases. Then run the code and show me the outputs.

*{attached code}* |
| Medium | Code Execution | Code to Code (Same language) | I wrote the following function which finds all of the composite numbers from 2 to 'n'. It's a bit slow though because it's a brute force method. Can you rewrite it in a way that reduces the time complexity? Use a few examples and time it to show the real differences in running time by executing the code.

*{attached code}* |
| Medium | Code Execution | Code generation and execution (Text to Code) | Create a Python decorator named measure_time that logs the duration a function takes to execute. Apply this decorator to a sorting function that sorts a list of random numbers. Test the sorting function with lists of increasing sizes (e.g., 100, 1,000, 10,000, and 100,000 elements) and use the decorator to print out the sorting times for each list size. Remember to run your code and give me the results. Make sure your code is clean and easy to follow! |
| Hard | Code Execution with File Uploads | Data Analysis | Can you please write a python file utilizing the library Streamlit for frontend visualization, and show case the amount of wine sales for each customer and display in a chart like diagram? Please sum "Amount", "Cash" and "Check" for each customer and set the value as "Gross Purchase". I attached a Winery_sales excel spreadsheet file in my prompt for you to follow. Please run your program and show me the visualization. Save the results.

*{attached file}* |
| Hard | Programming Assistant with Execution | Code Understanding | I have a code that simulates a game of Blackjack, but I'm not sure if it's accurate or efficient. The code takes into account the player's hand, the dealer's upcard, and the player's current bet. It returns the recommended action (hit, stand, double down, or split) based on a complex set of rules. However, I'm not sure if the rules are correctly implemented, and I'd like you to review the code and suggest improvements. Help me review the code, identify any issues, and suggest improvements. Then provide some test cases and run the improved code to demonstrate its accuracy.

*{attached code}* |
| Hard | Code Execution | Create example usages of this function | My friend has provided me with a game that is implemented as a Python function. I am not able to understand the code. Can you please explain to me how the game works by providing some example usages of this function. I want you to execute the program by providing some valid inputs. Please show me the outputs obtained and explain to me the reason for the outputs.

*{attached code}* |
| Hard | Code Execution | Code generation and execution (Text to Code) | Cryptarithms are like puzzles where you replace digits in a math equation with letters. Each letter stands for a unique digit. For example, in "SEND + MORE = MONEY", each letter is a unique digit. Please help me write a Python function that can crack any cryptarithm with addition, subtraction, multiplication, and division. The function should spit out a dictionary where each letter maps to the digit it represents. Please also provide some complex test cases and execute the programs you wrote to get the answers to these cases. |

Table 26: **Examples of the prompt set for tool use & coding capability.**

| Difficulty | L1 Category | L2 Category | Prompt |
|---|---|---|---|
| Easy | Scientific Reasoning | Chemistry | Given the reactants sodium chloride and silver nitrate, what are the products that will be formed? Give me the balanced chemical reaction. I don't care if the answer is generated or you get it with any tool, but please be as detailed as possible. |
| Easy | Mathematical Calculation | Complex Analysis | Show me the local minimum of function $f(x) = \frac{1}{3}x^3 - \frac{5}{2}x^2 + 4x$ |
| Medium | Scientific Reasoning | Engineering | A spacecraft is moving in gravity-free space along a straight path when its pilot decides to accelerate forward. He turns on the thrusters, and burned fuel is ejected at a constant rate of $2.0 \times 10^2$ kg/s at a speed (relative to the rocket) of $2.5 \times 10^2$ m/s. The initial mass of the spacecraft and its unburned fuel is $2.0 \times 10^4$ kg and the thrusters are on for 30 s. What is the thrust on the spacecraft and what is the acceleration? |
| Medium | Scientific Reasoning | Health and Medicine | Calculate the total calories burned by a person weighing 70 kg who walks at a speed of 5 km/h for 1 hour on a flat surface. Assume a walking metabolic equivalent (MET) value of 3.8. Use the formula for caloric expenditure: $$\text{Calories burned} = \text{MET} \times \text{weight in kg} \times \text{duration in hours}.$$ Additionally, factor in the effect of air resistance, assuming a wind speed of 10 km/h against the direction of walking, and adjust the MET value accordingly using the formula: $$\text{Adjusted MET} = \text{MET} \times \left(1 + 0.1 \times \frac{\text{wind speed in km/h}}{\text{walking speed}}\right).$$ Provide the detailed calculation steps and final result. |
| Hard | Mathematical Calculation | Ordinary and partial differential equations | Solve the first-order ordinary differential equation (ODE) using the Euler method: $$\frac{dy}{dx} = 3x + 2y, \text{ with the initial condition } y(0) = 1.$$ Calculate the solution from $x = 0$ to $x = 1$ with a step size of $h = 0.1$. Implement the Euler method to solve this ODE and print the value of $y$ at each step. Provide a Python function to represent the ODE and another to perform the Euler method. Give me the accurate results. |
| Hard | Mathematical Calculation | Plotting | Plot and shade in the region of the 2D coordinate plane defined by the inequality $|x| + y^2 < 10$. Show me the image, and save it png format. |
| Hard | Scientific Reasoning | Food Science | Could you suggest a one-day meal plan from McDonald's with meals that are roughly 400, 800, and 1200 calories each? Please include a drink with each meal and keep the calorie count within 20 calories of these targets. |
| Hard | Mathematical Calculation | Discrete math & logic | Given the logical statement $(r \rightarrow s) \wedge (t \rightarrow u)$ is true, and the statement $(s \wedge u) \rightarrow v$ is false, determine the truth value of the statement $$[\neg(r \wedge t) \vee (s \vee u)] \leftrightarrow [(r \vee t) \wedge (s \wedge u) \wedge \neg v].$$ Any tools are allowed. Explain your answer step by step. |

Table 27: **Examples of the prompt set for tool use & reasoning capability.**

## D.5 REFERENCE EXAMPLES

To offer a more intuitive understanding of the reference examples in our benchmark, we randomly select an instance from both *English* and *Reasoning* capabilities and present their reference examples in Tables 28 – 33.

| **Prompt** |
|---|
| **L1 Category:** Recommendations / Brainstorming
**L2 Category:** Entertainment suggestions
**Difficulty:** Hard
**Prompt:** I'm looking for suggestions of 1970's movies. The genre should be horror, and it should take place in the woods or be summer camp based. Do not include movies that also incorporate humor, the movies chosen need to be strictly horror. Include the titles and the directors. Also, include the names of any actors who have gone on to be successful in Hollywood after they were in these movies. |
| **Model Response 1** |
| Here are some 1970s horror movies that take place in the woods or at a summer camp, without humor:

1. **The Evil Dead** (1981) - Director: Sam Raimi
  - This film takes place in a remote cabin in the woods and follows a group of friends who are terrorized by demons. The movie has since become a cult classic.
  - Successful actors: Bruce Campbell, Ellen Sandweiss, and Richard DeManincor
2. **The Burning** (1981) - Director: Tony Maylam
  - This summer camp slasher film tells the story of a group of campers who are stalked and killed by a former camper who was horribly burned in a prank gone wrong.
  - Successful actors: Brian Matthews, Lou David, and Holly Hunter
3. **The Town That Dreaded Sundown** (1976) - Director: Charles B. Pierce
  - This horror film is based on a true story and takes place in a small town in Arkansas where a serial killer is terrorizing locals and tourists alike.
  - Successful actors: Dawn Wells, Jimmy Lydon, and Andrew Prine
4. **Just Before Dawn** (1981) - Director: Jeff Lieberman
  - This backwoods horror film follows a group of friends who are on a hiking trip in the Pacific Northwest when they are stalked and killed by a mysterious figure.
  - Successful actors: George Kennedy, Chris Lemmon, and Mike Randolph
5. **Madman** (1977) - Director: Joe Giannandrea
  - This summer camp slasher film tells the story of a group of campers who are stalked and killed by a legendary killer who is said to haunt the woods.
  - Successful actors: Paul Ehlers, Deborah Luce, and Thomas Carr
6. **The Prey** (1977) - Director: Edwin Brown
  - This wilderness survival horror film follows a group of friends who are stranded in the woods and hunted by a group of mountain men.
  - Successful actors: Keith Carradine, Tulsa McReynolds, and Lew Saunders
These movies are all considered horror classics and are still popular among fans of the genre today. |
| **Human Annotations** |
| **Rating 1:** 2/5
**Explanation 1:** This response makes several errors. The movie Madmen was directed by Joe Giannone, not Joe Giannandrea. It was also released in 1981, not 1977. The movie The Prey was released in 1983, not 1977. While the other movie dates are correct, most of them are not from the 1970s, which was a specific component of the user's request. The only movie in the suggested list that was actually released in the 1970s was The Town That Dreaded Sundown, which does not have the setting the user requested - summer camp or woods. |
| **Rating 2:** 2/5
**Explanation 2:** Half of its movie recommendations are from the 1980s. Since only 1970s movies were requested this leads to a lot of wasted recommendations. |

Table 28: **First reference example of English capability.** Each instance includes a prompt, 3 model responses, and 2 human scores with explanations. This table presents the first model response alongside its human annotations. Together, the model response and annotations serve as the reference example for the subsequent evaluation.

| Prompt |
|---|
| **L1 Category:** Recommendations / Brainstorming |
| **L2 Category:** Entertainment suggestions |
| **Difficulty:** Hard |
| **Prompt:** I'm looking for suggestions of 1970's movies. The genre should be horror, and it should take place in the woods or be summer camp based. Do not include movies that also incorporate humor, the movies chosen need to be strictly horror. Include the titles and the directors. Also, include the names of any actors who have gone on to be successful in Hollywood after they were in these movies. |
| **Model Response 2** |
| A great request! Here are some classic horror movies from the 1970s that fit your criteria: |

1. **The Last House on the Left (1972)** - Directed by Wes Craven
Starring: Sandra Peabody, Lucy Grantham, David Hess
A group of friends are stalked and murdered by a pair of escaped convicts in the woods.
2. **The Texas Chain Saw Massacre (1974)** - Directed by Tobe Hooper
Starring: Gunnar Hansen, Teri McMinn, Edwin Neal
A group of friends encounter a family of cannibals in rural Texas.
3. **The Hills Have Eyes (1977)** - Directed by Wes Craven
Starring: Susan Lanier, Robert Houston, Martin Speer
A family's trip to the desert turns into a nightmare when they encounter a family of mutants.
4. **Friday the 13th (1980, but filmed in 1979)** - Directed by Sean S. Cunningham
Starring: Betsy Palmer, Adrienne King, Kevin Bacon
A group of camp counselors are stalked and murdered by Jason Voorhees at a summer camp.
5. **The Town That Dreaded Sundown (1976)** - Directed by Charles B. Pierce
Starring: Ben Johnson, Andrew Prine, Dawn Wells
A small town is terrorized by a serial killer in this loosely based-on-true-events horror film.
6. **Madman (1978)** - Directed by Joe Giannandrea
Starring: Harriet Bass, Jimmy Steele, Seth Jones
A group of summer camp counselors are stalked and murdered by a legendary killer known as Madman Marz.

Notable actors who went on to achieve success in Hollywood after appearing in these films include:

- Kevin Bacon (Friday the 13th) - has had a successful career in film and TV, with roles in Footloose, Apollo 13, and American Crime Story.
- Gunnar Hansen (The Texas Chain Saw Massacre) - went on to appear in numerous horror films and write several books, including a memoir about his time on the set of The Texas Chain Saw Massacre.
- Wes Craven (The Last House on the Left, The Hills Have Eyes) - went on to become a legendary horror director, creating the Nightmare on Elm Street and Scream franchises.

Note: While some of these films may have been re-released or remade with added comedic elements, the original versions are strictly horror films.

| Human Annotations |
|---|
| **Rating 1:** 3/5 |
| **Explanation 1:** The list of movies is related to what the prompt is searching for but a lot of the details are off like years the movie was made and the names of directors. The prompt asks for 1970s genre horror, but a 1980 horror movie based In a summer camp like Friday the 13th would still be a useful result. The inaccuracies are a bit distracting and provide misinformation. |
| **Rating 2:** 3/5 |
| **Explanation 2:** This response does a great job of highlighting the horror actors who went on to greater fame. Most of the movies in the list were released in the 1970's, as requested by the user, and the list notes that one movie released in 1980 was filmed in the 1970's. However, the response states that Madman was directed by Joe Giannandrea and released in 1978, but both IMDB.com and Wikipedia note that this director's name is Joe Giannone and the movie was released in 1981. Additionally, The Town That Dreaded Sundown was not set in a summer camp, or in the woods, as requested in the prompt. |

Table 29: **Second reference example of English capability.** Each instance includes a prompt, 3 model responses, and 2 human scores with explanations. This table presents the second model response alongside its human annotations. Together, the model response and annotations serve as the reference example for the subsequent evaluation.

| Prompt |
|---|
| **L1 Category:** Recommendations / Brainstorming
**L2 Category:** Entertainment suggestions
**Difficulty:** Hard
**Prompt:** I'm looking for suggestions of 1970's movies. The genre should be horror, and it should take place in the woods or be summer camp based. Do not include movies that also incorporate humor, the movies chosen need to be strictly horror. Include the titles and the directors. Also, include the names of any actors who have gone on to be successful in Hollywood after they were in these movies. |
| **Model Response 3** |
| Sure, here are some horror movies from the 1970s that take place in the woods or involve summer camps, and they stick strictly to the horror genre:

1. **Title:** The Texas Chain Saw Massacre (1974)
 - **Director:** Tobe Hooper
 - **Notable Actors:** While the cast was relatively unknown at the time and didn't go on to become major Hollywood stars, the film itself has become a cornerstone of the horror genre.
2. **Title:** Deliverance (1972)
 - **Director:** John Boorman
 - **Notable Actors:** Burt Reynolds, Jon Voight
  - Jon Voight went on to have a successful career in Hollywood with films like Coming Home (for which he won an Academy Award) and Mission: Impossible.
3. **Title:** The Last House on the Left (1972)
 - **Director:** Wes Craven
 - **Notable Actors:** While no actors from this film went on to major Hollywood fame, the movie is notable for being Wes Craven's directorial debut, who later became a legendary horror director.
4. **Title:** The Hills Have Eyes (1977)
 - **Director:** Wes Craven
 - **Notable Actors:** Dee Wallace
  - Dee Wallace went on to star in films like E.T. the Extra-Terrestrial and Cujo.

While these films may not have large ensembles of widely recognized stars, they are significant in the horror genre and feature actors who have been successful in Hollywood. |
| **Human Annotations** |
| **Rating 1:** 4/5
**Explanation 1:** These are all 1970s horror movies that wouldn't be considered comedy. The movie year release dates, directors and actor information is accurate. The response could be improved with a summary of the plot. |
| **Rating 2:** 4/5
**Explanation 2:** This response provides specific details on the fame acquired by certain actors in these horror movies. The response only notes four movies, which seems small for this genre and decade, but they are all from the 1970's and 3 of the 4 are set in the woods or a summer camp. The Hills have Eyes is set in the desert, but it is a 'wilderness' setting. |

Table 30: **Third reference example of English capability.** Each instance includes a prompt, 3 model responses, and 2 human scores with explanations. This table presents the third model response alongside its human annotations. Together, the model response and annotations serve as the reference example for the subsequent evaluation.

| Prompt |
|---|
| **L1 Category:** Mathematical Calculation
**L2 Category:** Discrete math & logic
**Difficulty:** Medium
**Prompt:** Jane won the lottery and decided to spend some of the money. She spent $1.50 on the first day. She spent $3 on the second day. She spent $4.50 on the third day. She kept spending her winnings in the same pattern and then on the last day, she spent her remaining $300. How much did she win in the lottery? |
| **Model Response 1** |
| Jane spent $1.50 on the first day, $3 on the second day, and $4.50 on the third day. So that is 1.50 + 3 + 4.50 = 9 dollars in the first three days. She then spent $300 on the last day. So she spent 9 + 300 = 309 dollars in total. The answer is 309. |
| **Human Annotations** |
| **Rating 1:** 1/5
**Explanation 1:** The response does not understand that the user is asking for a pattern to be identified, and fails to understand that there were probably several days in between the third day and the final day. The response does not come up with a suitable formula and fails to correctly calculate the result; it instead just adds up the amounts specified in the prompt. |
| **Rating 2:** 2/5
**Explanation 2:** The response was incorrect because it failed to identify the pattern in the given logic. The prompt indicates that Jane spent money according to a pattern observed in the first three days, but the response only calculated the total money spent on the first three days and the last day, ignoring the expenditures on the middle days. |

Table 31: **First reference example of reasoning capability.** Each instance includes a prompt, 3 model responses, and 2 human scores with explanations. This table presents the first model response alongside its human annotations. Together, the model response and annotations serve as the reference example for the subsequent evaluation.

| Prompt |
| --- |

**L1 Category:** Mathematical Calculation
**L2 Category:** Discrete math & logic
**Difficulty:** Medium
**Prompt:** Jane won the lottery and decided to spend some of the money. She spent $1.50 on the first day. She spent $3 on the second day. She spent $4.50 on the third day. She kept spending her winnings in the same pattern and then on the last day, she spent her remaining $300. How much did she win in the lottery?

| Model Response 2 |
| --- |

On the first day, Jane spent 1.50 dollars. On the second day, she spent 3 dollars. On the third day, she spent 4.50 dollars. The pattern continues and the money she spends doubles each day. So on the 4th day, she spends 9 dollars, on the 5th day she spends 18 dollars, and so on. If we add up the money spent on each day, we will get a sequence of numbers that looks like this:

1.50, 3, 4.50, 9, 18, 36, 72, 144, 288, ...

Adding these numbers up, we get 576. Now we know that on the last day, she spent the rest of her money, which was 300 dollars. So the total money she spent is 576 + 300 = 876 dollars. The answer is 876.

| Human Annotations |
| --- |

**Rating 1:** 2/5
**Explanation 1:** The identified pattern was incorrect. The correct pattern was to add 1.5 on each subsequent day of spending, as observed in the first three days. The pattern in the response incorrectly suggested doubling the spending each day, which only applies correctly to the second day, not the first and third days.

**Rating 2:** 1/5
**Explanation 2:** The response knows it needs to find a pattern, but it is unable to find the correct one. Because the first to values in the series ($1.50 and $3) represent a doubling, it decides that this is the pattern, even though the next number in the series is $4.50. The fact that the pattern it identifies doesn't include $300 should indicate that the formula may not be correct, but instead the response just adds the $300 on at the end of its calculation. Its calculation fails because it is based on this faulty formula.

Table 32: **Second reference example of reasoning capability.** Each instance includes a prompt, 3 model responses, and 2 human scores with explanations. This table presents the second model response alongside its human annotations. Together, the model response and annotations serve as the reference example for the subsequent evaluation.

| Prompt |
| --- |
| **L1 Category:** Mathematical Calculation
**L2 Category:** Discrete math & logic
**Difficulty:** Medium
**Prompt:** Jane won the lottery and decided to spend some of the money. She spent $1.50 on the first day. She spent $3 on the second day. She spent $4.50 on the third day. She kept spending her winnings in the same pattern and then on the last day, she spent her remaining $300. How much did she win in the lottery? |

| Model Response 3 |
| --- |
| Let's analyze the situation:

Day 1: Jane spent $1.50
Day 2: Jane spent $3
Day 3: Jane spent $4.50

We can see that the amount she spent each day is increasing by $1.50. To find the total amount she spent before the last day, we need to find the number of days she spent money.

Let's assume she spent money for n days before the last day. Then, the total amount she spent before the last day is:

$1.50 + $3 + $4.50 + ... + $(1.5n)

This is an arithmetic series with first term $1.50, common difference $1.50, and n terms. The sum of the series is:

(1.5n)(n+1)/2

We know that the last term of the series is $1.5n. Since she spent $300 on the last day, we can set up the equation:

(1.5n)(n+1)/2 + $300 = Total amount won

We also know that the last term of the series is $1.5n = $300 - $1.50 (since she spent $1.50 on the first day). Solving for n, we get:

$1.5n = $298.50
n = 199

Now we can find the total amount she won:

(1.5(199))(199+1)/2 + $300 = Total amount won
(298.5)(200)/2 + $300 = Total amount won
$29850 + $300 = Total amount won
$30150 = Total amount won

The final answer is $30150. |

| Human Annotations |
| --- |
| **Rating 1:** 3/5
**Explanation 1:** The response identifies and describes the correct pattern as well as providing the correct answer ($30150). But the explanation of the formula it provides is very confusing, particularly with the way that it separates the $300 on the last day. It is not certain that the user would be able to make use of the formula provided, but at least they would know what the pattern is, after reading this response. |
| **Rating 2:** 5/5
**Explanation 2:** The response was correct because it identified the pattern of adding $1.5 on each subsequent day of spending. It correctly calculated the total amount spent before the last day using the appropriate formula and provided the accurate result of spending from the first day to the last day. |

Table 33: **Third reference example of reasoning capability.** Each instance includes a prompt, 3 model responses, and 2 human scores with explanations. This table presents the third model response alongside its human annotations. Together, the model response and annotations serve as the reference example for the subsequent evaluation.

### D.6 GUIDELINES FOR DIFFICULTY LEVELS

To illustrate the difficulty of prompts in our evaluations, below is a summary of the definitions, accompanied by examples, for easy, medium, and hard levels for the *English (multilingual)* and *Image Recognition* capabilities. We provide at least 3 examples to illustrate the distinctions and establish clear expectations for the different difficulty levels. They serve as instructional tools to help professional annotators from our data vendor better understand our difficulty level criteria, enabling them to annotate prompts of varying difficulty accurately.

#### D.6.1 ENGLISH AND MULTILINGUAL

**Easy**

**Definition** Prompt is a single ask/requirement/constraint for the model presented as a single statement **OR** prompt is a single statement without ask/requirement/constraints **AND** would not require subject matter expertise to understand.

**Examples**

- Illustrate and explain the proper use of a semi-colon.
- How do I uninvite my brother to my wedding?
- I've been having trouble sticking to my healthy diet lately. Give me some motivational words or tips to help me make better food choices and achieve my health goals.

**Medium**

**Definition** Prompt includes 2–4 asks/requirements/constraints for the model AND would not require subject matter expertise to produce a response.

**Examples**

- My neighbors blast loud music all night, and I can't sleep. I've tried talking to them directly, as well as calling 311 but nothing has changed. What else do you think I can try?
- How do I ask my boss for a raise? I think I'm underpaid but my boss never has time for me.
- Pretend you're Bugs Bunny. I'm Elmer Fudd. How would you greet me?
- Write me a funny haiku about dogs.

**Hard**

**Definition** Prompt contains 5 or more asks/requirements/constraints for the model OR requires subject matter expertise above and beyond "common knowledge" in order to respond.

**Examples**

- Write a poem to say sorry to my dog because I didn't spend enough time with it. The poem should have 26 lines where each line begins with Z, Y, X, ..., A, respectively, and always ends with h. The poem cannot contain any animal words.
- Sort the following words alphabetically, and in the result remove the first and the fourth words while capitalizing the rest: sioux fortescue purloin percept helmsman friend friends. Append a new lower-case word that is an animal living in Antarctica. Output the result with numbered bullets.
- Handling long-sequence inputs presents a significant challenge to the KV-cache of Transformers. Can we address this challenge better by training Transformers with more GPUs?
- I'm hosting a dinner party next week. I have a kosher friend coming, but also a vegan friend. Also, I am allergic to nuts. My husband likes spicy food. There might be a few picky eaters who are coming too. They may come with kids who attend preschools. What do you think I should make for dinner? And what about drinks?

### D.6.2   IMAGE RECOGNITION

**Easy**

**Definition**   Requires to stick to the image focus (instead of the background and details) **AND** requires NO external knowledge to answer the question (e.g. historical details, specific skills) **AND** requires NO fine-grained object recognition (e.g. plant species, aircraft models) **AND** requires NO complicated language format constraint (e.g. multi-level bullets, a specific order of listing, creative writing).

**Examples**

- How many pieces of chess are there? Please answer with one English word.
- What is the title of the presentation slides?
- What color is the watch belt?

**Medium**

**Definition**   Neither easy nor hard.

**Examples**

- Who are the cartoon characters in this image? What are they holding? (what they hold is not image focus)
- How do I fix this at home? (requires some knowledge)
- What is the species of this cute cat? (fine-grained recognition)

**Hard**

**Definition**   Requires identification of five or more entities, each possessing distinct characteristics or components, amid conditions of visual complexity. This includes scenarios where unrelated visual elements may interfere (visual distraction), or where relevant parts of the entities are partially obscured from view (visual occlusion), thus complicating the recognition process **OR** requires complicated format for language generation (e.g. multi-level bullets, a specific order of listing, creative writing) **OR** requires visual-related professional knowledge.

**Examples**

- Can you count the balloons of each color? (The image includes more than five balloons along with distractions like children, hats, and other objects.)
- It is said that there is a human face in this image. Can you explain how that can be? (comprehensive image understanding related to illusion)
- When was this photo taken? Can you tell me more about the related event? (visual recognition related to history knowledge)

The distinction between **Medium** and **Hard** difficulty hinges on whether the prompt necessitates visual-related professional knowledge. It's important to note that a prompt demanding professional knowledge does not automatically qualify as **Hard** unless that knowledge is specifically related to visual interpretation. For instance, if a prompt can be deconstructed into a part that solely concerns visual identification and another part that solely concerns factual knowledge, then the knowledge required is not considered visual-related.

## D.7 PROMPTS FOR EVALUATION

We provide the complete version of the system and evaluation prompts we adopt for LLM-as-a-Judge in Tables 34 and 35, respectively.

---

You are an expert AI evaluator tasked with assessing model responses. Rate the response using a 1-5 Likert scale according to the following rubrics:

### Rubrics:
  - **5/5 - Amazing:** The response is flawless and could hardly be improved.
  - **4/5 - Pretty Good:** The response is quite good, but has room for minor improvements.
  - **3/5 - Okay:** They are middle-of-the-road responses that could be improved in several ways.
  - **2/5 - Pretty Bad:** The response has major problems in helpfulness, truthfulness, or safety.
  - **1/5 - Horrible:** They are terrible responses and you would caution others against using models that generate responses like this.

Note: User prompts or model responses may include attachments. To ensure a thorough evaluation, you may need to write and execute code.

---

Table 34: **System prompt for LLM-as-a-Judge in our CROSSEVAL benchmark.**

[Attached]:
{*attached text*}

[User Prompt]:
{*user prompt*}:

To calibrate your evaluation, consider these reference examples:

[Reference Example 1]:
Model Response: {*response 1*}
Rating 1: {*rating 1*}/5 — Explanation 1: {*explanation 1*}
Rating 2: {*rating 2*}/5 — Explanation 2: {*explanation 2*}

[Reference Example 2]:
Model Response: {*response 2*}
Rating 1: {*rating 1*}/5 — Explanation 1: {*explanation 1*}
Rating 2: {*rating 2*}/5 — Explanation 2: {*explanation 2*}

[Reference Example 3]:
Model Response: {*response 3*}
Rating 1: {*rating 1*}/5 — Explanation 1: {*explanation 1*}
Rating 2: {*rating 2*}/5 — Explanation 2: {*explanation 2*}

**Use these examples as benchmarks for your evaluation scale and scoring consistency.** Here is the model response for evaluation:

[Model Response to be Evaluated]:
{*model response*}

Please provide your evaluation in the following format:

#### User Prompt Analysis
   - Identify key requirements and objectives from the user prompt.

#### Reference Examples Insights
   - Summarize scoring patterns and typical point deductions.
   - Include how many points should be deducted for each issue.

#### Model Response Evaluation
   - Pros: List strengths and positive aspects.
   - Cons: Identify weaknesses, **specifying point deductions for each**.

#### Holistic Assessment    - Consider if major strengths outweigh minor issues.
   - Combine similar deductions to avoid double penalization.
   - Balance deductions and positive aspects, and then explain your scoring decision.

#### Evaluation Score
Score: [X]/5

Ensure your evaluation is thorough, fair, and aligned with the reference examples. Your expertise is crucial in providing an accurate and insightful assessment.

Table 35: **Evaluation prompt for LLM-as-a-Judge in our CROSSEVAL benchmark.**

### D.8 MORE DISCUSSIONS ON LLM EVALUATOR

**Discussion on Tool Use.** In the benchmark, prompts related to tool use involve functionalities such as web browsing and code interpretation. However, the LLM APIs we experiment with do not support web browsing, and only the GPT-4 API supports code interpreters.

Fortunately, when we specify the date of the reference examples and indicate that the answers may be dynamic, LLMs without web browsing features can still serve as effective evaluators, achieving Pearson correlations above 0.6 across all tool use-related capabilities. Additionally, enabling GPT's code interpreter results in similar correlation scores but incurs higher costs. This may be because the reference examples already provide sufficient context for evaluation, eliminating the need for the model to execute code. As a result, we disable the code interpreter in subsequent evaluations.

**Final Evaluator Selection.** Table 1 shows that different LLMs excel at different capabilities. This naturally leads to the idea of using a mixture of LLMs as evaluators. For example, we could use Claude 3.5 to evaluate *Spanish & Image Recognition* and GPT-4o to evaluate *Spanish & Reasoning*, aiming for a higher overall correlation. However, this approach proves impractical due to significant differences in scoring distributions across models: Claude 3.5 tends to give higher scores, while GPT-4o is more stringent. While this discrepancy is not an issue when presenting a single score for the benchmark, it poses issues when analyzing the relationship between individual and cross-capability performance. The varying scoring distributions could make our conclusions unreliable. As a result, we select GPT-4o as our final evaluator, while providing the results using Claude 3.5 in the Appendix E.2 for reference.

### D.9 CASE STUDY FOR LLM-AS-A-JUDGE ON CROSSEVAL

We randomly select one instance from the CROSSEVAL benchmark to demonstrate the format and reliability of LLM-as-a-Judge in Tables 36 and 37, using GPT-4o and Claude 3.5 Sonnet as evaluators, respectively.

| Prompt |
| --- |
| The distance between cities X and Y is 240 miles. A train leaves from city X for city Z and another train leaves from city Y to City X simultaneously. It is known that one train is traveling 15 miles per hour faster than the other. After 3 hours, they find that they are 90 miles apart. Can we calculate the speeds of both trains? |

| Model Response from o1-preview |
| --- |

*The detailed calculation process is omitted for brevity.*

**Answer: Yes; their speeds are 17.5 mph and 32.5 mph—we can calculate both train speeds from the given data.**

| GPT-as-a-Judge |
| --- |

#### User Prompt Analysis
- The user prompt provides a scenario involving two trains traveling towards each other from different cities, with one train traveling faster than the other. The key requirements are to determine the speeds of both trains given the distance between the cities, the time traveled, and the distance apart after 3 hours.

#### Reference Examples Insights
- **Reference Example 1:** Deducted points for incorrect calculations and assumptions.
- **Reference Example 2:** Deducted points for incorrect assumptions about city Z and not addressing potential confusion.
- **Reference Example 3:** Deducted points for not solving the problem but correctly identifying the need for more information.

Typical point deductions:
- Incorrect calculations: 2-3 points.
- Incorrect assumptions or failure to address potential confusion: 1-2 points.
- Failure to solve the problem when solvable: 2-3 points.

#### Model Response Evaluation
**Pros:**
1. **Logical Setup**: The response correctly sets up the problem by assuming the trains are moving towards each other along the line between cities X and Y.
2. **Correct Calculations**: The response accurately calculates the combined distance covered and the combined speed.
3. **Clear Explanation**: The steps are clearly explained, making it easy to follow the logic and calculations.

**Cons:**
1. **Assumption Clarification**: The response assumes that city Z is on the same line as cities X and Y without explicitly addressing the potential confusion in the prompt. This could lead to a minor deduction for not clarifying the assumption (1 point).

#### Holistic Assessment
- The response is strong in its logical setup, clear explanation, and correct calculations. The only minor issue is the lack of explicit clarification about the assumption regarding city Z. However, this does not significantly detract from the overall quality of the response.

#### Evaluation Score
Score: 4/5

The response is quite good, with only a minor area for improvement in clarifying assumptions. The major strengths outweigh this minor issue, leading to a high score.

Table 36: **Case study of GPT-as-a-Judge.**

| Prompt |
| --- |

**Capability:** Reasoning
**L1 Category:** Mathematical Reasoning
**L2 Category:** Math word problem solving
**Difficulty:** Hard
**Prompt:** The distance between cities X and Y is 240 miles. A train leaves from city X for city Z and another train leaves from city Y to City X simultaneously. It is known that one train is traveling 15 miles per hour faster than the other. After 3 hours, they find that they are 90 miles apart. Can we calculate the speeds of both trains?

| Model Response from o1-preview |
| --- |

To determine the speeds of both trains, let's reconsider the setup carefully. The distance between cities X and Y is 240 miles. A train leaves from city X for city Z, and another train leaves from city Y towards city X simultaneously. Since they are on the same rail line between cities X and Y, it's logical to assume that both trains are moving towards each other along that line.

*The detailed calculation process is omitted for brevity.*

**Answer: Yes; their speeds are 17.5 mph and 32.5 mph—we can calculate both train speeds from the given data.**

| Claude-as-a-Judge |
| --- |

#### User Prompt Analysis
The user prompt requires calculating the speeds of two trains given:
  - Distance between cities X and Y (240 miles)
  - One train leaves X for Z, another leaves Y for X
  - One train is 15 mph faster than the other
  - After 3 hours, they are 90 miles apart

Key objective: Determine if it's possible to calculate both train speeds with the given information.

#### Reference Examples Insights
  - Major point deductions (2-3 points) for failing to recognize the ambiguity in city Z's location
  - Minor deductions (1 point) for calculation errors or misinterpretation of given data
  - Full points for recognizing the problem cannot be solved without additional information
  - Partial credit for correct calculations if assuming a linear path between X and Y

#### Model Response Evaluation
Pros:
  - Correctly identifies the ambiguity in the original prompt regarding city Z
  - Assumes a linear path between X and Y, which is a reasonable interpretation
  - Calculations are mathematically correct given this assumption
  - Clearly explains the step-by-step process

Cons:
  - Fails to explicitly state that the original problem is unsolvable without more information about city Z (-1 point)
  - Doesn't acknowledge that the assumption of a linear path is an interpretation, not a given fact (-0.5 points)

#### Holistic Assessment
The model response demonstrates a good understanding of the problem and provides a mathematically sound solution based on a reasonable interpretation. While it doesn't explicitly address the ambiguity in the original prompt, it does implicitly recognize it by making an assumption about the train paths. The calculations and reasoning are clear and correct.

The main weakness is the lack of explicit acknowledgment of the original problem's ambiguity. However, the strength of the mathematical reasoning and the clarity of the explanation partially compensate for this oversight.

#### Evaluation Score
Score: 3.5/5

This score reflects the model's strong mathematical approach and clear explanation, balanced against the lack of explicit discussion about the original problem's ambiguity. It's a solid response that could be improved by directly addressing the limitations of the given information.

Table 37: **Case study of Claude-as-a-Judge.**

# E  "LAW OF THE WEAKEST LINK" OF CROSS CAPABILITIES IN LLMs

## E.1  EXPERIMENTAL SETUP

To ensure comprehensive coverage of LLM performance across capabilities, we select 17 models from five major model families: GPT (OpenAI, 2023), Claude (Anthropic, 2024), Gemini (Reid et al., 2024), Llama (Llama Team, 2024), and Reka (Ormazabal et al., 2024). Each model supports at least five cross-capability scenarios in our experiments (except o1 models). For consistency, we use the GPT-4o-05-13 model as the evaluator, with temperature set to 0 and seed set to 42 to ensure deterministic scoring. Each model's responses are generated using their default decoding parameters to achieve optimal performance. For the Llama 3.1 405B model, we specifically use the FP8 version. A complete list of model versions is provided Table 38. In addition, while the Gemini API supports code interpreter functionality, it does not yet handle non-text outputs (e.g., data plots), so we exclude its results on tool-use-related prompts in our benchmark.

| LLM Name | Model Version |
|---|---|
| GPT-4o-mini | gpt-4o-mini-2024-07-18 |
| GPT-4o | gpt-4o-2024-05-13 |
| o1-mini | o1-mini-2024-09-12 |
| o1-preview | o1-preview-2024-09-12 |
| Claude 3 Haiku | claude-3-haiku-20240307 |
| Claude 3 Sonnet | claude-3-sonnet-20240229 |
| Claude 3 Opus | claude-3-opus-20240229 |
| Claude 3.5 Sonnet | claude-3-5-sonnet-20240620 |
| Gemini 1.5 Flash | gemini-1.5-flash-001 |
| Gemini 1.5 Pro | gemini-1.5-pro-001 |
| Gemini 1.5 Pro Exp | gemini-1.5-pro-exp-0801 |
| Reka Edge | reka-edge-20240208 |
| Reka Flash | reka-flash-20240722 |
| Reka Core | reka-core-20240722 |
| Llama 8B | Llama 3.1 8B |
| Llama 70B | Llama 3.1 70B |
| Llama 405B | Llama 3.1 405B FP8 |

Table 38: **Exact LLMs versions used in our experiment.**

## E.2  RESULTS FOR CLAUDE-AS-A-JUDGE

To avoid potential bias from using a single evaluator, we present all results with Claude 3.5 Sonnet as the evaluator in Table 39. Notably, for the *Coding & Reasoning* task, the performance of five models falls between the weak and strong capabilities but tends to be closer to the strong one, as highlighted in purple in the Table. This may be due to the fact that *Coding* and *Reasoning* are key capabilities in current LLM development, with potentially many cross-capability prompts included in the training data, boosting LLM performance for this specific task. While this pattern does not appear in the GPT-as-a-Judge results, Claude-as-a-Judge still generally demonstrates a clear "Law of the Weakest Link" effect, as illustrated by the distribution in Figure 3.

| Individual Capabilities | | | | | | |
|---|---|---|---|---|---|---|
| **Models** | **English** | **Reasoning** | **Coding** | **Image** | **Tool Use** | **Long Context** | **Spanish** |
| Claude 3 Haiku | 64.92 | 58.17 | 67.08 | 56.60 | – | 70.80 | 66.03 |
| Claude 3 Sonnet | 72.48 | 64.36 | 73.12 | 62.75 | – | 72.03 | 69.99 |
| Claude 3 Opus | 73.17 | 69.37 | 74.74 | 65.42 | – | 73.77 | 75.86 |
| Claude 3.5 Sonnet | 78.22 | 76.52 | 77.37 | 77.70 | – | 76.60 | 76.03 |
| GPT-4o mini | 76.13 | 68.74 | 75.81 | 68.51 | – | 78.20 | 76.18 |
| GPT-4o | 78.60 | 74.69 | 76.43 | **77.35** | – | 83.48 | 80.63 |
| o1-mini | 77.28 | 84.28 | **87.01** | – | – | 83.39 | 83.80 |
| o1-preview | **82.63** | **88.85** | 86.49 | – | – | **86.70** | **86.24** |
| Gemini 1.5 Flash | 70.62 | 65.83 | 73.79 | 56.56 | – | 77.44 | 72.28 |
| Gemini 1.5 Pro | 75.93 | 75.14 | 75.19 | 73.86 | – | 79.32 | 77.34 |
| Gemini 1.5 Pro Exp | 77.42 | 75.61 | 75.62 | 76.67 | – | 80.11 | 80.87 |
| Reka Edge | 51.86 | 44.07 | 43.87 | 53.41 | – | 35.46 | 53.31 |
| Reka Flash | 65.29 | 62.36 | 64.37 | 61.14 | – | 53.22 | 70.60 |
| Reka Core | 73.77 | 72.44 | 70.14 | 60.21 | – | 62.69 | 74.24 |
| Llama 3.1 8B | 67.11 | 55.26 | 67.02 | – | 47.22 | 65.29 | 60.15 |
| Llama 3.1 70B | 71.82 | 64.46 | 71.66 | – | 48.33 | 67.59 | 64.92 |
| Llama 3.1 405B | 74.76 | 71.04 | 75.51 | – | **50.38** | 72.81 | 73.89 |

| Cross Capabilities | | | | | | |
|---|---|---|---|---|---|---|
| **Models** | **Coding & Rea.** | **Image & Rea.** | **Long & Coding** | **Spanish & Rea.** | **Spanish & Image** | **Tool & Coding** | **Tool & Rea.** |
| Claude 3 Haiku | 66.03 | 56.38 | 65.85 | 59.29 | 58.73 | – | – |
| Claude 3 Sonnet | 70.19 | 60.52 | 67.27 | 61.81 | 65.85 | – | – |
| Claude 3 Opus | 70.32 | 59.94 | 68.65 | 71.67 | 65.61 | – | – |
| Claude 3.5 Sonnet | 78.60 | 77.92 | 74.31 | 76.12 | 79.25 | – | – |
| GPT-4o mini | 74.82 | 68.27 | 71.31 | 69.42 | 67.83 | – | – |
| GPT-4o | 75.19 | **78.79** | 73.64 | 76.28 | **78.08** | 48.27 | **58.21** |
| o1-mini | 85.89 | – | 83.48 | 83.29 | – | – | – |
| o1-preview | **87.38** | – | **83.74** | **86.25** | – | – | – |
| Gemini 1.5 Flash | 71.54 | 54.71 | 69.31 | 68.82 | 53.66 | – | – |
| Gemini 1.5 Pro | 76.13 | 73.25 | 74.01 | 72.60 | 67.24 | – | – |
| Gemini 1.5 Pro Exp | 73.15 | 76.29 | 72.79 | 74.51 | 73.01 | – | – |
| Reka Edge | 47.63 | 30.64 | 23.03 | 39.18 | 43.88 | – | – |
| Reka Flash | 64.67 | 47.47 | 45.25 | 62.01 | 57.23 | – | – |
| Reka Core | 69.83 | 50.87 | 50.00 | 68.99 | 56.56 | – | – |
| Llama 3.1 8B | 61.39 | – | 55.26 | 47.65 | – | 51.49 | 50.17 |
| Llama 3.1 70B | 71.65 | – | 60.09 | 58.61 | – | 51.49 | 51.99 |
| Llama 3.1 405B | 71.85 | – | 64.42 | 67.40 | – | **56.13** | 57.03 |

Table 39: **Experimental results for individual and cross capabilities on CROSSEVAL using Claude as the evaluator.** To avoid potential evaluator bias, we present Claude's results solely as a reference point and bold the best non-Claude results. In cross-capability evaluations, we define one of the involved individual capabilities as stronger and the other as weaker if the absolute score difference between them exceeds $\Delta = 3$ points. In 48 cross-capability scenarios where this difference is present, 24 cases show performance lower than both individual capabilities ( red background ), 18 show performance between the two but closer to the weaker capability ( blue background ), and 6 show performance closer to the stronger capability ( purple background ). Notably, no cross-capability score ever exceeds the stronger individual capability.

### E.3 DISCUSSION ON DISTINGUISHING "WEAK" AND "STRONG" CAPABILITIES

In the experiments presented in the main text, we identify "strong" and "weak" capabilities within cross-capability tasks when the absolute difference between their individual scores exceeds $\Delta = 3$. To illustrate the effect of $\Delta$ on "Law of the Weakest Link," we adjust its value from 1 to 6 and plot the density distribution using GPT-4o and Claude 3.5 Sonnet as evaluators, as shown in Figures 4 and 5. Notably, regardless of the chosen $\Delta$ value, cross-capability performance consistently clusters around the weaker performance, clearly demonstrating the "Law of the Weakest Link."

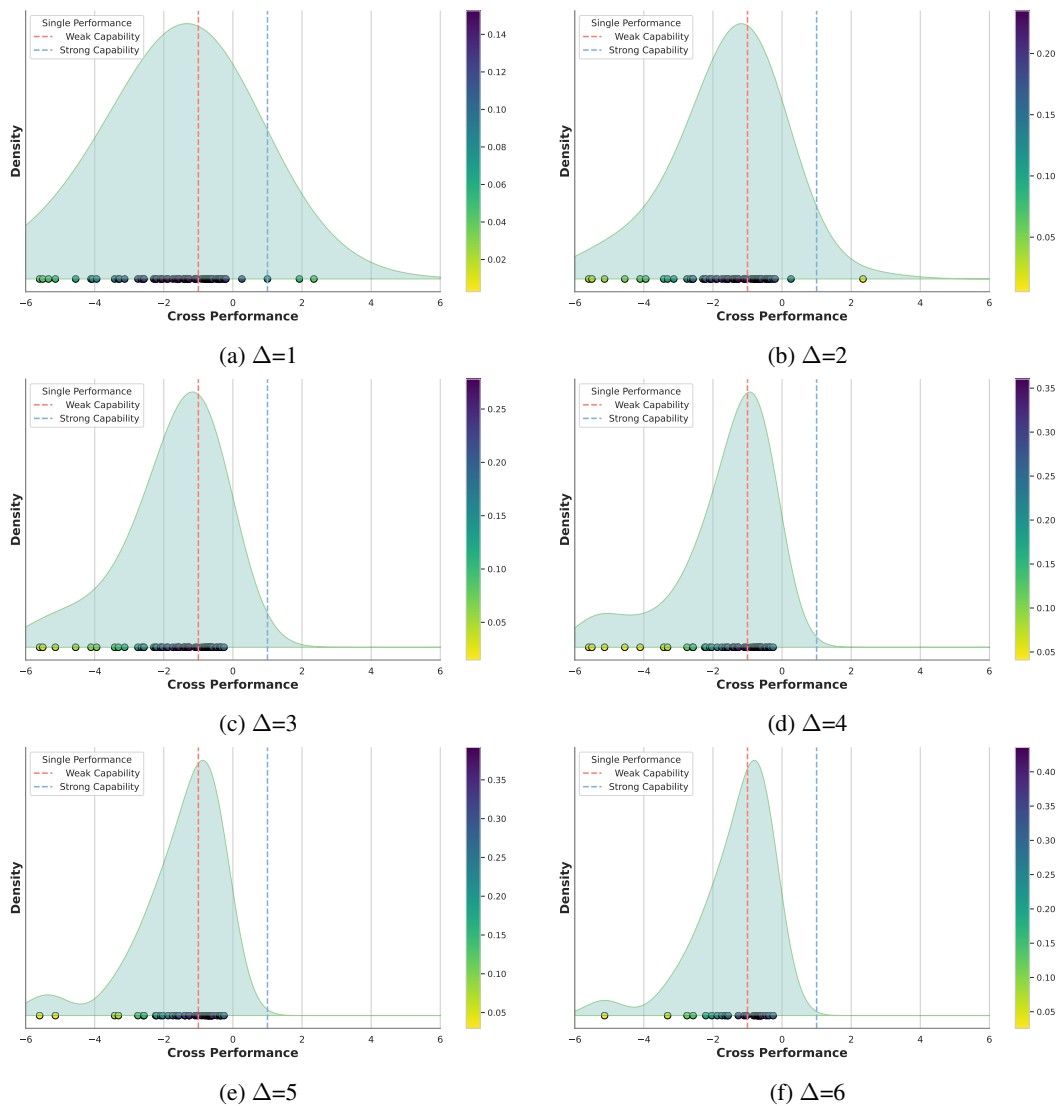

Figure 4: **Effect of $\Delta$ on the density distribution of cross-capability performance evaluated by GPT-4o.**

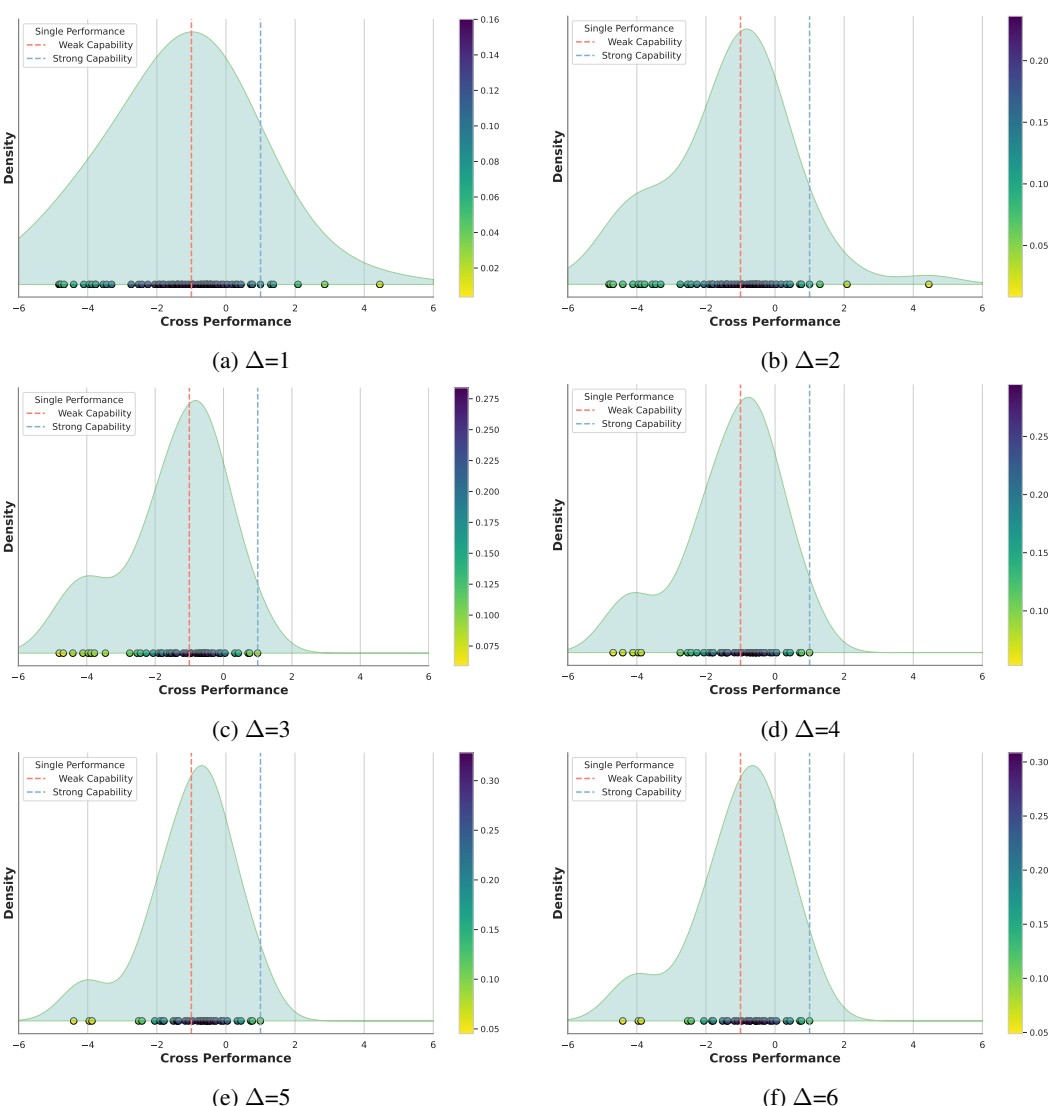

Figure 5: **Effect of $\Delta$ on the density distribution of cross-capability performance evaluated by Claude 3.5 Sonnet.**

### E.4 RESULTS FOR DIFFERENT DIFFICULTY LEVELS

In Table 40, we present the scores of 17 models across prompt sets with varying levels of difficulty. It's important to note that these scores are not directly comparable across different model families, as they support varying capabilities. For instance, while Llama does not support *Image Recognition*, it covers all capabilities related to *Tool Use*.

Nevertheless, as shown in Table 40, 12 out of the 17 models perform better on the Easy prompt set compared to the Medium set, and similarly, they score higher on the Medium set than the Hard set. This pattern suggests that the difficulty levels we manually defined align well with model performance. An exception to this trend is the Claude model family, where all four Claude models scored slightly higher on the Hard prompt set than on the Medium set.

| Models | Easy | Medium | Hard |
|---|---|---|---|
| GPT-4o mini | 75.14 | 69.51 | 69.00 |
| GPT-4o | 76.87 | 69.82 | 68.97 |
| o1-mini | 82.52 | 78.63 | 77.75 |
| o1-preview | 84.07 | 79.25 | 78.07 |
| Claude 3 Haiku | 64.23 | 57.85 | 58.58 |
| Claude 3 Sonnet | 68.16 | 61.41 | 62.86 |
| Claude 3 Opus | 72.21 | 64.41 | 65.17 |
| Claude 3.5 Sonnet | 76.94 | 70.86 | 70.91 |
| Gemini 1.5 Flash | 71.65 | 62.28 | 61.86 |
| Gemini 1.5 Pro | 76.60 | 68.26 | 69.44 |
| Gemini 1.5 Pro Exp | 79.86 | 71.20 | 70.08 |
| Reka Edge | 45.89 | 40.71 | 40.43 |
| Reka Flash | 60.74 | 55.86 | 55.57 |
| Reka Core | 69.85 | 60.69 | 59.32 |
| Llama 3.1 8B | 55.00 | 52.34 | 50.69 |
| Llama 3.1 70B | 64.00 | 61.06 | 57.94 |
| Llama 3.1 405B | 70.41 | 62.63 | 61.77 |

Table 40: **Results of different difficulty levels evaluated by GPT-4o.**

# F  INVESTIGATING INDIVIDUAL-CAPABILITY ALTERATIONS

## F.1  PROMPT TO GENERATE PRINCIPLE

The complete prompt used for automatically generating principles is provided in Table 41.

---

You are an AI expert tasked with analyzing common mistakes in model responses and creating a comprehensive set of principles to improve the {*capability*} of the model. We will work step-by-step to build this guideline. Specifically, for each iteration, I will provide you with one instance, and you need to update the current principles accordingly. There are 100 instances in total, and the principles should be completed after reviewing all instances.

For each instance, you have the following information:
   - User Prompt
   - Model Response
   - Evaluation of the Model Response
   - Current Principles

**### Instance** {*index*}
{*current instance, including the user prompt, model response, and evaluation using an LLM-as-a-judge*}

For each iteration, choose **ONE of the following actions:**
1. **ADD**
   - Introduce a new principle that isn't currently listed.
2. **REPLACE**
   - Replace a less significant principle with a new one.
   - Clearly specify which principle is being replaced.
3. **REVISE**
   - Enhance the principles by making them more detailed and specific.
4. **KEEP**
   - If the current instance is already covered by existing principles, leave the guideline unchanged.

**Current Principles:**
{*current principles*}:

**Output Format:**
**## Summary**
   - Summarize any major issues with the present response.
   - Provide specific, actionable steps to prevent these errors, if any.
   - Based on your summary and the current principles, decide which action (ADD, REPLACE, REVISE, or KEEP) should be taken for the current instance.

**## Principles for Prompts related to** {*capability*}
**### Principle 1**: Title [Use the title to specify the context in which this principle should be applied, such as "For Legal Reasoning" or "For Mathematical Reasoning"]
   - Include up to three key points.
   - Each point should be directly applicable to the model's generation process without requiring additional training or resources.
   - Each point must be extremely specific to allow for direct execution.
     - For example, instead of saying "Use a structured markdown format," clearly define the exact format for each step, including the structure for the beginning, middle, and end.
     - Instead of advising to "avoid vague terms," provide a specific list of terms to be avoided.
     - Rather than generally suggesting "avoid errors in math calculations" or "double-check," outline concrete steps to prevent such errors.
...

[END of Principles]

**Requirements:**
   - Follow the output format exactly, including "[END of Principles]" at the end with no remarks after it.
   - Include up to 10 distinct principles in the report. If there are already 10 principles, "ADD" is not allowed.
   - You may reorder the principles as necessary: Place important, typical, and representative principles at the front, while less important ones can be moved toward the back.
   - **Ensure that each suggestion in the principles is detailed and actionable, rather than being a general description.**

---

Table 41: **Prompt for generating principles based on responses from CROSSEVAL.**

## F.2 CASE STUDY FOR PRINCIPLE-BASED SYSTEM PROMPTS

Using Gemini 1.5 Flash as an example, we present the automatically generated system prompts for the *Reasoning* capability in Tables 42 – 44. The "Note" in Table 44 is added manually.

---

## Principles for Prompts related to Reasoning

### Principle 1: For Mathematical Reasoning
- **Verify All Mathematical Steps and Properties Thoroughly**:
   1. Validate each step in mathematical derivation meticulously, focusing on crucial values, properties, and boundary conditions.
   2. Ensure consistency in the use of all variables and constants across the steps.
   3. Verify the accuracy of factorization and simplification steps.
- **Detail Intermediate Calculations and Logical Steps Clearly**:
   1. Show all steps in complex calculations for transparency and clarity.
   2. Justify intermediate steps thoroughly, explicitly stating relevant formulas and boundary terms.
- **Identify and Correct Misleading Statements and Errors**:
   1. Scrutinize claims about variable independence, solution behavior, or mathematical properties for accuracy.
   2. Correct misinterpretations about connectivity or group properties for given spaces.
- **Complete Execution of Methods**:
   1. Ensure the final steps of methods like finding steady-state vectors are executed.
   2. Provide exact values and solutions without leaving the explanation incomplete.
- **Balance Thoroughness and Conciseness**:
   1. Ensure comprehensive coverage of essential steps without over-explaining or redundancy.
   2. Avoid unnecessary repetition and condense explanations suitably while maintaining clarity.
- **Leverage Degree Mismatches**:
   1. Use polynomial degree comparisons to simplify analysis, especially when identifying potential function inverses.
   2. Note that two polynomials of different degrees cannot be inverses of each other.

### Principle 2: For Proving Statements Involving the Pigeonhole Principle
- **Explicitly State the Contradiction**:
   1. Clearly state why having all elements unique causes a contradiction in the problem's context.
   2. Provide specific examples if necessary to illustrate the contradiction.
- **Outline Logical Assumptions**:
   1. Clearly state every logical assumption at the start of the proof.
   2. Reiterate these assumptions when reaching a conclusion to reinforce the logic.
- **Detail the Application of Principles**:
   1. Clearly detail how and why the pigeonhole principle is applied, linking each step back to the problem's context.

### Principle 3: For Logical Sequencing and Step-by-Step Explanations
- **Detail Logical Steps Clearly**:
   1. Ensure each step is explained in detail, showing how one leads to the next.
   2. Break down complex proofs or problems into components, revealing the underlying reasoning.
- **Explicitly Address and Evaluate Assumptions**:
   1. Clearly state assumptions made and justify their relevance to the problem.
   2. Evaluate each assumption for feasibility and update reasoning if new information is revealed.
   3. Explain how these assumptions influence conclusions drawn.
- **Incorporate Relevant Legal and Logical Principles**:
   1. Include specific legal principles or doctrines if applicable.
   2. Explain these principles in context and link to the problem's scenario.
- **Ensure Accurate Initial Dependency Analysis**:
   1. Validate initial dependencies comprehensively before analyzing post-observation changes.
- **Maintain Logical Cohesion**:
   1. Ensure explanation maintains a logical flow from start to finish.
   2. Avoid ambiguities, ensuring each point connects clearly to the next.
- **Comprehensive Coverage**: Address all potential dependencies and independencies to ensure logical completeness.

---

Table 42: **System prompt (Principles 1-3) automatically generated to enhance the reasoning capability of Gemini 1.5 Flash.**

## Principles for Prompts related to Reasoning

### Principle 4: For Addressing Ambiguities and Considering Multiple Possibilities
- **Identify and Resolve Ambiguities**:
    1. Point out any ambiguous terms or conditions within the problem statement.
    2. Clearly state how these ambiguities are resolved.
- **Make Assumptions Clear**:
    1. Articulate any assumptions made to proceed with the solution.
    2. Justify why these assumptions are reasonable and how they influence results.
- **Evaluate All Possible Correct Answers**:
    1. Ensure that all potential correct answers are considered and evaluated.
    2. If multiple answers are possible, acknowledge them explicitly and explain why each is plausible.
- **Re-Evaluate Intermediate Assumptions**:
    1. Consistently check interim assumptions for feasibility as the solution progresses.
    2. Correct initial assumptions if they fail to align with further logical deductions.

### Principle 5: For Financial Analysis and Reasoning
- **Step-by-Step Financial Calculations**:
    1. Break down financial calculations into detailed, transparent steps.
    2. Show intermediate steps clearly, not just the final results.
- **Compare Different Scenarios**:
    1. Provide comparisons of different financial scenarios when applicable.
- **Highlight Key Conclusions**:
    1. Summarize key financial implications explicitly.
- **Tailor Negotiation Strategies**:
    1. Provide negotiation tactics specific to each buyer's unique offer.
    2. Include concrete phrases or tactics the user can use.
    3. Justify financial recommendations clearly within the user's context.

### Principle 6: For Hypothesis Development in Scientific Contexts
- **Ensure Comprehensive Factor Coverage**:
    1. Verify all specific factors mentioned in the prompt are addressed.
- **Avoid Redundancy**:
    1. Consolidate related points to prevent repetition.
- **Provide Clear Hypotheses**:
    1. Present hypotheses clearly and in testable terms.

### Principle 7: For Scientific Reasoning and Empirical Analysis
- **Verify the Existence of Citations**:
    1. Confirm all citations are based on actual research papers, cross-referencing with recognized academic databases.
    2. Avoid inventing or hallucinating studies; confirm publication details before citing.
- **Summarize Study Findings Accurately**:
    1. Provide specific results and data points from studies to back claims.
    2. Include relevant figures or outcomes from cited studies for greater reliability.
- **Incorporate Empirical Evidence**:
    1. Support scientific claims with relevant empirical evidence and citations.
    2. Avoid overgeneralizations; use specific examples or case studies.

Table 43: **System prompt (Principles 4-7) automatically generated to enhance the reasoning capability of Gemini 1.5 Flash.**

## Principles for Prompts related to Reasoning

### Principle 8: For Designing Scientific Experiments
- **Detail the Measurement Methods**:
    1. Specify tools and procedures for measuring each variable.
    2. Include details like frequency of measurements and exact techniques used.
- **Clarify Statistical Analysis**:
    1. Explain how statistical tests will analyze collected data.
    2. Provide details on data preparation and results interpretation.
- **Verify Citations**:
    1. Ensure all literature references are verifiable and credible.
    2. Cross-reference cited studies with recognized academic databases.

### Principle 9: For Summarizing and Analyzing Policies
- **Incorporate Procedural Details**:
    1. Include specific procedural elements like roles and responsibilities.
    2. Enhance comparisons with explicit distinctions and summarize key differences and similarities.
    3. Incorporate references to relevant cases or statutes.
- **Avoid Redundancy**:
    1. Consolidate related information to avoid repetition.
- **Include Interpretative Analysis**:
    1. Interpret how regulations impact the environment they govern.
    2. Clarify the rationale or feasibility of suggested legal arguments.
- **Address All Policy Elements**:
    1. Summarize all major sections, including scope, administration, restrictions, and enforcement.

### Principle 10: For Ethical Reasoning
- **Avoid Redundancy**:
    1. Consolidate related ethical advice.
- **Incorporate Ethical Theories**:
    1. Explicitly mention ethical theories like consequentialism, deontology, and virtue ethics.
- **Focus on Specific Actionable Steps**:
    1. Provide detailed steps for addressing ethical issues.

**Note**:
    - Apply the principles above to generate better responses for user prompts that require reasoning.
    - For prompts that do not require reasoning, disregard these principles.
    - Avoid quoting or referencing these principles, as the user is not aware of its existence.

[END of Reasoning Principles]

Table 44: **System prompt (Principles 8-10) automatically generated to enhance the reasoning capability of Gemini 1.5 Flash.**

