# OpenReview forum: "Law of the Weakest Link: Cross Capabilities of Large Language Models"
_ICLR.cc/2025/Conference — ICLR 2025 Poster_

### Official Review · Reviewer_zpqw · 2024-10-25

**Soundness:** 3
**Presentation:** 4
**Contribution:** 3
**Rating:** 8
**Confidence:** 3

**Summary:**

This paper introduces the concept of cross capabilities in LLMs. It defines seven individual and seven cross capabilities, creating a taxonomy to evaluate LLM performance. The authors introduce CROSSEVAL, a benchmark with 1,400 prompts to test both individual and cross capabilities. Results reveal that LLMs often follow the Law of the Weakest Link, where the weakest capability limits performance in cross-capability tasks.

**Strengths:**

(1) The paper provides a detailed taxonomy of individual and cross capabilities in LLMs.

(2) The CROSSEVAL benchmark is a good contribution. It consists of 1,400 human-annotated prompts that rigorously test LLMs on individual and cross capabilities.

(3) The paper identifies a critical pattern in LLM performance -- the Law of the Weakest Link. This result and finding are insightful.

(4) The paper is well-written. And code and data are shared.

**Weaknesses:**

(1) Although the paper includes a multilingual aspect (with Spanish), the evaluation of cross capabilities involving languages other than English is limited.

(2) The study concentrates on evaluating capabilities in pairs, but it does not investigate scenarios requiring the integration of more than two capabilities simultaneously.

(3) Need a detailed analysis of specific errors LLMs make in cross-capability tasks. A breakdown of common failures for particular capability combinations could provide more insights.

**Questions:**

Can the authors clarify the selection criteria for annotators and their domain expertise? How did the authors address potential biases in human ratings and ensure the reliability of the annotations?

---

> ### Author Response · Authors · 2024-11-22
> **Author Response (Part 1)**
>
> We extend our thanks for the careful consideration and valuable insights you've shared. Our answers to your questions are presented as follows.
>
> > **W1: Although the paper includes a multilingual aspect (with Spanish), the evaluation of cross capabilities involving languages other than English is limited.**
>
> Thank you for pointing this out. We acknowledge that the coverage of multilingual capabilities in this paper is not extensive. Incorporating additional cross capabilities involving other languages significantly increases the expertise required from annotators (e.g., Spanish & Reasoning). This poses a challenge in sourcing a sufficient number of qualified professional annotators, even through data vendors.
>
> Therefore, we adopt the same approach as [1], selecting Spanish as a representative language for multilingual capabilities. Accordingly, our analysis focuses specifically on the interactions of Spanish with other capabilities, such as Spanish & Reasoning and Spanish & Image Recognition. We do not make broader claims about the intersection of multilingualism as a whole with these capabilities.
>
> We hope this explanation clarifies our rationale and the scope of our analysis.
>
> > **W2: The study concentrates on evaluating capabilities in pairs, but it does not investigate scenarios requiring the integration of more than two capabilities simultaneously.**
>
> Thank you for your insightful suggestion. As this is the first systematic evaluation and investigation of cross capabilities, we intentionally limit the analysis to pairs of capabilities. This focused approach serves several purposes:
>
> - It establishes a clear foundation for defining and understanding cross capabilities in a structured manner.
> - It facilitates comprehensive analysis of the relationship between individual and cross capabilities without introducing unnecessary complexity.
> - It ensures rigorous and manageable evaluations, particularly given the significant challenges in sourcing qualified annotators and creating high-quality benchmarks for tasks involving more than two capabilities.
>
> We fully agree that investigating scenarios requiring the integration of more than two capabilities is an important direction for future research, building on the insights and methodologies established in this work.
>
> > **W3: Need a detailed analysis of specific errors LLMs make in cross-capability tasks. A breakdown of common failures for particular capability combinations could provide more insights.**
>
> Thank you for your constructive feedback. While we do not include a manual error analysis, we address this in Section 5 by proposing a principle-based prompting method to automatically summarize common failure modes and provide actionable principles to avoid such errors for specified capabilities.
>
> We agree that a detailed analysis of model errors across capabilities could offer valuable insights to the community. To support this, we plan to release all the generated analysis presented in Section 5. Additionally, we will make publicly available the responses of 17 models across 14 capabilities, along with the LLM-based evaluations. We hope this will enable future research to explore manual or more advanced automated methods for analyzing model failures in both individual and cross-capability tasks.

---

> ### Author Response · Authors · 2024-11-22
> **Author Response (Part 2)**
>
> > **Q1: Can the authors clarify the selection criteria for annotators and their domain expertise? How did the authors address potential biases in human ratings and ensure the reliability of the annotations?**
>
> We provide the following clarifications regarding the selection criteria for annotators, their domain expertise, and the steps we take to address potential biases and ensure reliability in the annotations:
>
> **Selection of Annotators.** We first establish an internal team of research scientists and product managers with LLM experience (including some of the authors of this paper). This team designs the required capabilities, their taxonomy, annotation guidelines, and difficulty level guidelines, which are then shared with our data vendor.
>    - The data vendor employs hundreds of professional annotators with expertise in various domains, such as native Spanish speakers or individuals skilled in reasoning or coding.
>    - Annotators are selected by the data vendor based on the specific requirements of each capability.
>
> **Ensuring Reliability of Annotations.** To ensure high-quality annotations:
>    - Our internal team conducts several rounds of discussions to draft and refine all taxonomies and guidelines before handing them over to the data vendor.
>    - After the initial annotations are completed, our team reviews the results, iteratively requesting updates from the annotators when necessary.
>    - This iterative process helps maintain the highest possible reliability and consistency in the annotations.
>
> **Addressing Potential Biases in Human Ratings.** As noted in Lines 235–251 of the paper, evaluating LLM responses is inherently subjective and challenging, even for professional annotators. To mitigate potential biases, we take the following steps:
>    - Conduct three rounds of pilot annotations on 30% of the benchmark, continuously refining the guidelines and collaborating with the data vendor to align scoring criteria.
>    - Utilize a diverse pool of annotators from various backgrounds and expertise areas across 14 different capabilities, which significantly reduces the likelihood of systemic biases.
>    - Demonstrate in the results that our process achieves a substantial level of agreement across the entire benchmark.
>
> We hope the above addresses your questions. If you have any remaining concerns, we would be happy to discuss and resolve them further.
>
> ---
>
> **References:**
>
> [1] SEAL LLM Leaderboards. https://scale.com/leaderboard.

---

> > ### Comment · Reviewer_zpqw · 2024-11-24
> >
> > Thanks for the clarifications. I will keep my original positive rating.

---

> > > ### Author Response · Authors · 2024-11-26
> > > **Thank you for your review**
> > >
> > > Thank you for taking the time to read our response! We would like to express our gratitude once more for your detailed review and constructive feedback.

---

### Official Review · Reviewer_Gjoy · 2024-11-03

**Soundness:** 3
**Presentation:** 3
**Contribution:** 3
**Rating:** 6
**Confidence:** 3

**Summary:**

This paper investigates cross-capabilities - the intersection of multiple distinct abilities across different types of expertise needed to solve complex, real-world tasks. The authors comprehensively define individual and cross-capabilities in LLMs and carefully annotate prompts to evaluate these cross-capabilities. Their findings reveal that current LLMs follow the "Law of the Weakest Link," where cross-capability performance is significantly limited by the weakest component capability. Additionally, they discovered that improving the weakest capabilities leads to the greatest improvements in LLMs' general capabilities.

**Strengths:**

- The investigation of cross-capabilities is both interesting and important.
- The prompt collection and annotation methodology is comprehensive and reliable.
- The insights about the "Law of the Weakest Link" and the finding that "improving the weakest capabilities leads to the greatest improvements" are particularly valuable.

**Weaknesses:**

- This work focuses more on benchmarking rather than algorithm design. It would be more appropriately categorized as a data/benchmark paper.
- The study's scale is somewhat limited, with a dataset of only 1,400 prompts.

**Questions:**

Could you provide more explanation about:
- The rationale for incorporating reference responses in Section 3.3.2
- How this relates to correlation between capabilities?

---

> ### Author Response · Authors · 2024-11-22
> **Author Response**
>
> We are grateful for your valuable review and thoughtful suggestions, and we try to answer your questions as follows.
>
> > **W1: This work focuses more on benchmarking rather than algorithm design. It would be more appropriately categorized as a data/benchmark paper.**
>
> Thank you for the suggestion. We would like to clarify that our paper is indeed submitted under the "Datasets and Benchmarks" area. For ICLR, "Datasets and Benchmarks" is one of the primary areas, not a separate submission track.
>
> > **W2: The study's scale is somewhat limited, with a dataset of only 1,400 prompts.**
>
> Thank you for your feedback. We would like to clarify the following points regarding the dataset size:
>
> - **Comparison to contemporary studies:** Our dataset, with 1,400 prompts, is already the largest of its kind compared to other contemporary works. For example, [1] includes 765 instances, and [2] contains 1,024 instances. Additionally, while these works often rely on significant assistance from LLMs during the annotation process [1] or use existing conversation logs [2], our benchmark is annotated entirely from scratch by professional annotators. This process adheres strictly to a rigorous taxonomy and detailed guidelines developed by a team of research scientists and product managers.
>
> - **Annotation efforts beyond prompts:** Beyond the 1,400 prompts, we include 3 reference responses and 6 human ratings with detailed explanations for each prompt, resulting in a total of 8,400 human reviews to ensure reliable LLM-based evaluations.
>
> Given the scale of prompts, the quality and depth of annotation, and the extensive human reviews, we believe our benchmark, designed to evaluate open-ended responses across various capabilities, represents the largest and most robust dataset of its kind. We hope this clarification demonstrates that the dataset size is a strength of our work rather than a limitation.
>
> > **Q1: Could you provide more explanation about: The rationale for incorporating reference responses in Section 3.3.2 & How this relates to correlation between capabilities?**
>
> Apologies for any confusion caused by Section 3.3.2. We address your questions as follows:
>
> - **Rationale for incorporating reference responses:** As mentioned in the paper, our benchmark can not annotate ground truth directly. To address this, we include multiple reference examples to simulate the role of ground truth in the evaluation process. Notably, we include not only reference responses but also the corresponding human ratings with explanations for each response. This setup allows the model to align with consistent scoring standards, common deduction criteria, and potentially correct answers derived from human explanations when evaluating new responses. Our experiments demonstrate that increasing the number of reference examples improves the reliability of LLM-based evaluations, resulting in stronger correlations with human ratings.
>
> - **Relation to correlation between capabilities:** Reference examples are designed to improve the reliability of evaluations for individual instances within a single capability by simulating ground truth. They do not influence or alter the "correlation between capabilities." Their role is confined to enhancing the evaluation process for the current capability being assessed.
>
> We hope this clarification addresses your concerns. However, if we have misunderstood your question, please feel free to provide further details, and we would be happy to elaborate.
>
> ---
>
> **References:**
>
> [1] Kim et al. The BiGGen Bench: A Principled Benchmark for Fine-grained Evaluation of Language Models with Language Models.
>
> [2] Lin et al. WildBench: Benchmarking LLMs with Challenging Tasks from Real Users in the Wild.

---

> > ### Comment · Reviewer_Gjoy · 2024-11-28
> > **Reply to the rebuttal**
> >
> > Thank you for the reply. Most of my concerns have been addressed. And I prefer to keep my current score.

---

> ### Author Response · Authors · 2024-11-28
> **Thank you for your review**
>
> Thank you for taking the time to review our response! We sincerely appreciate your thoughtful feedback and are delighted to hear that we are able to address your concerns.

---

### Official Review · Reviewer_A1B6 · 2024-11-03

**Soundness:** 2
**Presentation:** 3
**Contribution:** 3
**Rating:** 6
**Confidence:** 2

**Summary:**

This paper constructs a benchmark for the core and cross capabilities. It identifies seven core capabilities and proposes seven cross-capabilities based on them. The author proposes CROSSEVAL to evaluate each capability. Throughout the extensive experiment, the author claims the Law of the Weakrdy link, meaning that the cross capability is heavily influenced by the weakest core capability associated with it.

**Strengths:**

1. The topic of this paper appears to be interesting. A proper taxonomy of the ability of LLM is necessary.
2. The paper writing is good. The whole paper flow is easy to follow.

**Weaknesses:**

1. The reviewer fails to understand why constructing prompts for core capabilities. Don't there exist many benchmarks for specific core capabilities?
2. The reviewer doesn't understand why each capability is comparable to the others, considering the difference in prompts.

**Questions:**

1. What are the criteria for selecting core capabilities and cross capabilities? It appears the these capabilities are somehow randomly selected without proper justification. For instance, why are *English* and *Spanish* selected as core capabilities like *long-context* instead of creating a different dimension named *lingual*?
2. Won't this paper be more suitable for a benchmark track instead of a research paper?
3. How is the image encoded in LLM? Does the LLM include other modalities?

---

> ### Author Response · Authors · 2024-11-22
> **Author Response (Part 1)**
>
> Thanks for the detailed review and helpful comments. In response to your questions, our answers are as follows.
>
> > **W1: The reviewer fails to understand why constructing prompts for core capabilities. Don't there exist many benchmarks for specific core capabilities?**
>
> We apologize for not elaborating on this in the paper. While existing benchmarks for specific core capabilities exist, we find it necessary to construct prompts for the following reasons:
>
> 1. **Lack of Clear Categorization:** Existing benchmarks often lack precise categorization for individual prompts. For example, benchmarks evaluating coding tasks may include prompts that, according to our taxonomy, fall under either the individual capability "Coding" or the cross-capability "Coding & Reasoning." Without this categorization, it is impossible to conduct the in-depth analyses needed to understand the relationships between individual and cross capabilities.
>
> 2. **Coverage:** Existing benchmarks do not cover all 76 Level-1 and 332 Level-2 categories in our taxonomy. To ensure thorough evaluation, we construct prompts to fill these gaps and provide more complete coverage.
>
> 3. **Difficulty Distribution:** Existing benchmarks often lack consistent difficulty distribution or use varying distributions across tasks. These inconsistencies hinder systematic comparisons of individual and cross capabilities. Our benchmark standardizes difficulty levels across all capabilities, supporting meaningful analysis.
>
> 4. **Style Distribution:** Existing benchmarks vary significantly in prompt styles, such as QA formats, exam-like questions, or real-world scenarios. These variations hinder consistency in evaluation. To address this, we standardize all prompts in our benchmark to reflect open-ended, real-world user queries, ensuring uniformity across capabilities.
>
> We hope this explanation clarifies why constructing our benchmark is essential for systematically studying cross-capability interactions and advancing the evaluation of LLM performance.
>
> > **W2: The reviewer doesn't understand why each capability is comparable to the others, considering the difference in prompts.**
>
> Thank you for raising this point about the comparability of different capabilities. We understand the concern and provide the following clarifications:
>
> 1. **Comparability of cross capabilities and individual capabilities:** In our analysis, when comparing individual capabilities with cross capabilities, the cross capabilities always include the individual capability as a component. For example, "Coding & Reasoning" incorporates "Coding" within its tasks, ensuring significant overlap in task types. This overlap makes the comparison between individual and cross capabilities meaningful, as it allows us to measure how well individual capabilities generalize and integrate with others.
>
> 2. **Consistency in difficulty and style:** As mentioned previously, we strictly control the difficulty and style distribution of prompts across all capabilities. By standardizing these factors, we ensure that comparisons are valid.
>
> 3. **No comparisons of unrelated capabilities:** We do not compare unrelated capabilities with no overlap, as we agree that such comparisons would lack significance. Our evaluations focus on scenarios where meaningful relationships exist, such as cross capabilities that naturally combine individual capabilities used in tandem for real-world tasks.
>
> 4. **Robust findings:** Across all evaluated scenarios, involving 14 capabilities and 17 models, our results consistently demonstrate the "Law of the Weakest Link." This robust and recurring observation reinforces the significance of our comparisons, revealing systematic trends in model performance and interactions between individual and cross capabilities.
>
> We hope this explanation demonstrates how we ensure meaningful and fair comparisons across capabilities.

---

> ### Author Response · Authors · 2024-11-22
> **Author Response (Part 2)**
>
> > **Q1: What are the criteria for selecting core capabilities and cross capabilities? It appears the these capabilities are somehow randomly selected without proper justification. For instance, why are English and Spanish selected as core capabilities like long-context instead of creating a different dimension named lingual?**
>
> Thank you for pointing out that we missed elaborating on this in the paper. We first provide clarifications regarding multilingualism:
>
> **Clarification on multilingualism.** In terms of multilingualism, we maintain consistency with previous research [1-4] in treating English and other languages as distinct capabilities:
>
> For English:
> - We align with the Llama 3 paper [1], categorizing tasks such as knowledge-based QA and instruction following under general English.
> - English is treated as a distinct capability representing the foundational attributes of English-centric LLMs. Notably, it does not engage in cross-capability scenarios in our framework.
>
> For other languages, there is currently no consensus on whether they should be counted as one or more capabilities:
> - [1, 2] treat all non-English languages as a single capability.
> - [3] evaluates each language (e.g., Chinese, French, German) as an individual capability.
>
> In our work, we adopt the approach used in [4], selecting Spanish as a representative language to study cross capabilities. Notably, our cross-capability scenarios focus specifically on Spanish's interaction with other capabilities (e.g., Spanish & Reasoning, Spanish & Image Recognition), without making broader claims about the intersection of multilingualism with these capabilities.
>
> **Criteria for Selecting Capabilities in Our Benchmark.** Regarding the selection of each capability in our benchmark, the criteria are as follows:
>
> - **Selection of Individual Capabilities:** We primarily follow the capability evaluations established for post-trained LLMs in the Llama 3 paper (see Table 16 in [1]), which represent capabilities of primary interest to the LLM community. Additionally, we include Image Recognition to reflect the growing multimodal capabilities of state-of-the-art LLMs. This ensures that our findings generalize across both language and vision modalities.
>
> - **Selection of Cross Capabilities:** To select cross-capability scenarios, we ensure the coverage of all individual capabilities and analyze the frequency of capability pairs based on public real user queries [5, 6]. This analysis allows us to identify the seven most common and practically relevant cross-capability scenarios.
>
> We appreciate your feedback and hope the explanation clarifies our methodology.
>
> > **Q2: Won't this paper be more suitable for a benchmark track instead of a research paper?**
>
> Thank you for the suggestion. We would like to clarify that our paper is indeed submitted under the "Datasets and Benchmarks" area. For ICLR, "Datasets and Benchmarks" is one of the primary areas, not a separate submission track.
>
> > **Q3: How is the image encoded in LLM? Does the LLM include other modalities?**
>
> Apologies for any misunderstanding. All the LLMs evaluated in this paper, except for Llama 3.1, are multimodal models that support at least text and image inputs. This marks a notable progression in the capabilities of LLMs, which initially focused exclusively on text but have since expanded to handle other modalities such as images. Despite these advancements, by convention, they are still referred to as LLMs, reflecting their core focus on language understanding.
>
> We understand the interest in how images are encoded in these models. However, as the state-of-the-art LLMs we evaluate are proprietary, we regret that we cannot provide specific technical details about their image-encoding processes. In this paper, through evaluating image-related individual and cross capabilities, we also aim to analyze phenomena that emerge across different modalities.
>
> ---
>
> **Please note that all detailed explanations regarding the issues you mentioned have been updated in the new version of the manuscript.** If you still have any concerns, please let us know. We would be happy to address them further and update our paper accordingly.
>
> ---
>
> **References:**
>
> [1] Llama Team. The Llama 3 Herd of Models.
>
> [2] Kim et al. The BiGGen Bench: A Principled Benchmark for Fine-grained Evaluation of Language Models with Language Models
>
> [3] Chatbot Arena LLM Leaderboard. https://lmarena.ai.
>
> [4] SEAL LLM Leaderboards. https://scale.com/leaderboard.
>
> [5] Zheng et al. LMSYS-Chat-1M: A Large-Scale Real-World LLM Conversation Dataset. ICLR 2024.
>
> [6] Zhao et al. WildChat: 1M ChatGPT Interaction Logs in the Wild. ICLR 2024.

---

> > ### Author Response · Authors · 2024-11-30
> > **Looking forward to the discussion**
> >
> > Dear Reviewer A1B6,
> >
> > Thank you for dedicating your time and effort to reviewing our work. We understand that your schedule may be quite busy and truly appreciate your thoughtful evaluation. As the author-reviewer discussion phase approaches its end, we kindly request your feedback on whether our response has effectively addressed your concerns and if there are any additional questions or points you would like to discuss.
> >
> > Looking forward to further discussion and your valuable insights.
> >
> > Best regards,
> >
> > The Authors

---

> > > ### Comment · Reviewer_A1B6 · 2024-12-02
> > >
> > > Thanks to the author for the detailed rebuttal. My concerns about W1, Q2 and Q3 are addressed. However, I still have the following concerns:
> > > - For W2, what I really mean is the difficulties are hard to control, regardless of your effort. Just like you can not directly compare the result of CIFAR10 and Fashion-MNIST on the image foundation models. You may compare the rank of different models on different tasks, reflecting on their relative ability. But directly comparing the accuracy does not appear to be reasonable. Specifically, the reviewer thinks it is reasonable to say LLM A is better than LLM B in long context understanding. But it is hard to say LLM A is better at long context understanding than at image encoding.
> > > - For Q1, the reviewer appreciates the author for their effort in presenting a prototype on the capability of LLM, the reviewer suggests the author should adopt a better framework to depict the LLM's capability (such as multi-modality) and these specific capabilities (such as image encoding) can be viewed as representatives. Additionally, the reviewer thinks it is better to bring separate laws for the pure language model and multi-modality-supported language models, respectively. It is better to clarify the easy tasks before diving into more difficult ones.

---

> > > > ### Author Response · Authors · 2024-12-02
> > > > **Discussions on W2 and Q1**
> > > >
> > > > Thank you for your thoughtful response! We are glad that some of your concerns, particularly W1, Q2, and Q3, have been addressed. For the remaining concerns, we would like to provide the following clarifications:
> > > >
> > > > > **Regarding W2**
> > > >
> > > > We completely agree that difficulty control is subjective and challenging. To address this, we have made extensive efforts, as detailed below:
> > > >
> > > > **Relevance of the compared capabilities.**
> > > > - First, we would like to clarify that *we never compare different individual capabilities,* as such comparisons, as you mentioned, would neither be meaningful nor reasonable. Instead, our comparisons focus on an individual capability and its corresponding cross capabilities. For example, we compare `coding` with `reasoning & coding`, where `coding` is a subset of `reasoning & coding`. Each prompt in the `reasoning & coding` contains specific tasks from `coding` taxonomy. Given their significant overlap, we believe such comparisons are valid.
> > > > - Regarding your example, we do not compare long context understanding with image encoding. Instead, we analyze individual capability and its corresponding cross capabilities that share overlapping task types, ensuring their comparability.
> > > >
> > > > **Dataset construction and difficulty control.**
> > > > - While CIFAR10 and Fashion-MNIST belong to the same image classification task, they differ significantly in image types and annotation standards. To address similar concerns, we adopt unified annotation guidelines, employ a consistent pool of professional annotators (e.g., the same pool of domain experts for coding-related tasks), and share detailed difficulty-level guidelines between individual and cross-capabilities. This ensures significant overlap in task types and careful alignment of prompts and difficulty levels, which differentiates our approach from CIFAR10 and Fashion-MNIST.
> > > > - Additionally, recognizing that perfect alignment is impossible, we further examine Spanish-related cross capabilities derived as direct translations of English prompts in individual capabilities. In these cases, the prompts and difficulty levels are completely aligned except for the language difference, yet we still observe a significant "Law of the Weakest Link" (Table 2).
> > > >
> > > > **Metrics used.**
> > > > - Another clarification is that we do not use accuracy as a metric. Instead, we rely on a 1-5 Likert scale, where 1 represents a horrible response and 5 represents a perfect one. We employ consistent rubrics and evaluation guidelines across all capabilities, ensuring that the resulting scores are comparable.
> > > >
> > > > We hope this clarifies the robustness and validity of our approach to comparing individual capabilities and their corresponding cross capabilities.
> > > >
> > > > > **Regarding Q1**
> > > >
> > > > Thank you for your suggestion! We fully agree with your perspective and acknowledge that the conclusions of this paper primarily focus on pure-text scenarios (e.g., 6 out of 7 individual capabilities and 5 out of 7 cross-capabilities are text-only). In line with this, our title and conclusions specifically refer to LLMs rather than vision-language models.
> > > >
> > > > Given the increasing trend of SOTA LLMs supporting multimodal inputs (e.g., Llama 3.2 now supports image inputs), we include image recognition as a representative multimodal capability to explore whether it aligns with the identified "Law of the Weakest Link." We will follow your recommendation and further clarify in the paper that this serves as a reference for currently multimodal-capable LLMs, rather than a comprehensive conclusion about vision-language models.
> > > >
> > > > ---
> > > >
> > > > We hope these clarifications address your remaining concerns and welcome any further discussion. Thank you once again for your valuable feedback!

---

> > > > > ### Comment · Reviewer_A1B6 · 2024-12-02
> > > > >
> > > > > Thanks for the instant reply. Generally, this paper addresses an interesting yet unexplored topic in the community. The majority of my concerns are addressed. I will raise my score to 6 and favour acceptance. Please see the below comments:
> > > > > - I still have doubts about W2, in that the shift of validation dataset can not be directly compared, even if it is between related capabilities such as *reasoning & coding* VS. *coding*. It would be better to present more justification on the comparable issue. For instance, the consistency of human labels between capabilities? I suggest the author add the above rebuttal as the limitation of this work.
> > > > > - About Q1, I suggest the author restructure the paper and provide clear and strong takeaways on both kinds of LLM.
> > > > > - [*Just Personal feelings*] Given the overall score distribution, I think the paper will have a high chance of being accepted. In that case, it is safe for the author to revise the paper in a more beneficial, inspiring and reflective tone rather than a defensive one. Sadly, the community is in no shortage of the latter and really needs the former these days ...

---

> > > > > > ### Author Response · Authors · 2024-12-02
> > > > > > **Thank you for your suggestions**
> > > > > >
> > > > > > We truly appreciate your thoughtful feedback and suggestions, which have greatly contributed to improving our paper.
> > > > > >
> > > > > > We apologize for not being able to incorporate these insights into the current manuscript as the update deadline has passed. However, we will certainly include them in our next version. Specifically, we plan to add a limitations section to discuss issues such as the lack of comprehensive multimodal capability categorization and the challenges of comparing different capabilities.
> > > > > >
> > > > > > We also deeply resonate with your personal feelings. This is why, in our conclusions, we specifically focus on Spanish rather than broadly claiming multilingual applicability, and highlight image recognition as a reference rather than making generalized statements about multimodal capabilities. In the next version, we will aim to clarify these points more precisely to ensure the reliability of the takeaways, while also emphasizing potential limitations to provide a balanced and reflective perspective.
> > > > > >
> > > > > > Thank you once again for your valuable feedback and support!

---

### Official Review · Reviewer_7cFP · 2024-11-03

**Soundness:** 3
**Presentation:** 4
**Contribution:** 2
**Rating:** 5
**Confidence:** 4

**Summary:**

The goal of the paper is to investigate how LLMs perform at the intersection of various capabilities. The paper starts off by remarking that most of the LLM evaluations tend to focus on single capabilities like long context reasoning and factual knowledge retrieval. In practice, however, LLMs may be used to perform tasks that leverage several such capabilities simultaneously. The main contribution of the paper is a meticulously contracted, human-annotated dataset. The dataset consists of prompts that aim to test one or more capabilities. In addition to the input prompts, the paper also asks humans to annotate responses by various SoTA LLMs. Noting the relatively low degree of inter-annotator agreement, the paper improves the annotation guidelines to increase the agreement. Next, the paper design effective prompting strategies for using LLMs as evaluators. The results of the experiments are quite interesting: the performance of the LLMs on compound capabilities correlates with the performance on the individual capabilities.

**Strengths:**

1. The paper is quite correct in pointing out that benchmarks have not kept up with the actual usage of LLMs in the wild. With that in mind, a systematic approach for studying the cross-capability performance of the models will be very much appreciated by the LLM community.

2. The paper is well-written and quite easy to follow. The paper provides illustrative examples at pretty much every step.

3. The human surveys are quite thorough. The authors spent significant effort in understanding the results and improving the annotation performance.

4. The model-based evaluation protocol in Section 3.3 is very meticulously designed. The protocol exposes many interesting choices (though some are from existing papers) which will be enhance readers' knowledge on conducting effective model based evals.

**Weaknesses:**

While the paper is a pleasure to read, and generally well-executed, I feel there are two main issues that still stand in the way of acceptance, namely, lack of precise definitions and lack of important details.

## Lack of precise definitions

It is very difficult to understand what counts as a capability, and the paper doesn't not discuss it in sufficient detail. How should a capability be theoretically defined? Are different MMLU categories, e.g., math, medicine, individual capabilities? Is sitting in the SAT exam a single capability? It does require a bunch of skills like recalling meaning of words and basic math formulas, analytical reasoning and equation solving. Having a precise definition of a capability seems quite important in understanding how generalizable the insights of the paper are.

The reader is left asking why certain skills are designated as different capabilities and where really the boundaries are. For instance, why are English and Spanish two different capabilities? Will all languages be different capabilities? What if the languages are closely related to each other, e.g., Hindi & Urdu, German & Dutch?

On a related note, do we assume that the capabilities are equally different from each other? For instance, coding and Image Recognition (different input modalities) seems much more different than English vs. Spanish (same modality, even some overlap of words and grammar). Is there a notion of capability Gerrymandering (akin to [fairness Gerrymandering](https://arxiv.org/abs/1711.05144)) where one could form cross capabilities in various ways to show completely different results?

## Missing details

The paper seems to be missing some critical details which are important to assessing the generalizability of the results.

1. How were the capabilities selected? What was left out? For instance, why not consider Long Context and Language?

2. Also, how are the capabilities broken down into Levels 1 and 2? How exhaustive is this breakdown?

3. How were the examples for annotators generated (Appendix B.4)? Could these example prime the annotators? If these examples are not from the benchmark (as the text notes), what is their propose?

4. Line 201: "selecting a leaf node from our taxonomy to categorize each prompt by capability". Where do these prompts come from?

5. How much coverage do these prompts provide of each capability?

6. Line 211: There are 100 to 500 annotated prompts for each capability, but then why are the reviewers asked to select a precise 100 examples out of it? Is it possible that some capabilities have better quality annotations than others because different sets of prompts were forced to be downsampled to a round number of 100? Why not set a fixed quality criteria and let all the prompts through even if the final number if higher / lower than 100?


### Small nits

3. Line 156: Why is “Brainstorming about Local and Current Things” a part of the Tool Use? Could you provide examples of tools that perform brainstorming?

**Questions:**

Please see the questions under the "Weaknesses heading".

---

> ### Author Response · Authors · 2024-11-22
> **Author Response (Part 1)**
>
> We appreciate your thorough review as well as constructive feedback, and try to answer your questions as follows.
>
> > **W1.1: It is very difficult to understand what counts as a capability, and the paper doesn't discuss it in sufficient detail.**
>
> We thank the reviewer for raising this insightful question regarding the definition and scope of "capabilities." To address this systematically, we offer the principles and definitions of each capability, compare our efforts to prior work, and explain how specific scenarios like MMLU or SAT exams fit within our framework.
>
> **Definition of capability.** In our framework, a capability is defined by the following principles:
>
> - *Distinctiveness*: The capability should have unique characteristics that distinguish it from others.
> - *Measurability*: The capability should be independently evaluatable through specific tasks.
> - *Consistency*: The capability should maintain its fundamental characteristics across different contexts and domains.
>
> Using these principles, we define the seven individual capabilities as follows:
>
> - *English:* The ability to understand, generate, and manipulate English text for tasks such as factual answering, procedural instructions, creative writing, and dialogue.
> - *Reasoning:* The ability to perform logical deduction, mathematical computation, analytical problem-solving, and domain-specific reasoning, including areas such as scientific, legal, and ethical contexts.
> - *Coding:* The ability to write, synthesize, document, debug, and review code across various programming languages.
> - *Image Recognition:* The ability to perceive, identify, interpret, and describe visual information.
> - *Tool Use:* The ability to interact with external tools and APIs, including functions such as web browsing, code execution, and file uploads.
> - *Long Context:* The ability to process, comprehend, and synthesize information to generate responses from extensive textual inputs, ranging from 16K to 128K tokens.
> - *Spanish:* The same as the English capability, but for the Spanish language.
>
> **Comparison with prior work on defining capabilities.** Existing benchmarks or leaderboards focus on evaluating the individual capabilities of LLMs but lack a structured framework to define and distinguish them [1-4]. Our work advances this by:
>
> - For cross capability: Being the first to propose the investigation of cross-capabilities, supported by benchmarks and in-depth analysis.
> - For individual capability: Introducing a structured framework to better understand individual capabilities, including definitions, detailed taxonomies, and difficulty level guidelines.
>
> While defining capabilities remains an evolving challenge in the LLM era, we hope our framework serves as a foundational step for further exploration and refinement within the community.
>
> **Categorization of standardized tests.** We do not classify entire exams (e.g., MMLU, SAT) as individual or cross capabilities. Instead, we categorize each question within the exam based on the specific capabilities it tests. For example:
>
> - A straightforward knowledge question about historical facts would fall under "English" capability.
> - A mathematical proof would fall under "Reasoning" capability.
> - SAT chart interpretation questions require "Image Recognition" (to understand visual data) and "Reasoning" (to analyze trends and draw conclusions).
>
> By categorizing each prompt/question at this granular level, we hope to ensure precise and interpretable evaluation aligned with real-world applications.

---

> ### Author Response · Authors · 2024-11-22
> **Author Response (Part 2)**
>
> > **W1.2: The reader is left asking why certain skills are designated as different capabilities and where really the boundaries are & questions about different languages.**
>
> Here are our clarifications and further explanations regarding the issues you raised about the relationship between skills and capabilities and the treatment of different languages:
>
> **Distinction between capability and skill.** The concept of "skill" is relatively vague and can manifest in different ways:
>
> - It can be a subset of a capability, fitting within our taxonomy (e.g., "basic math formulas" and "analytical reasoning" as part of Reasoning capability).
> - It may span multiple capabilities. For instance, “research” can be a skill that may require various capabilities: Tool Use (to gather information via web browsing), Long Context (to process and synthesize extensive documents), Reasoning (to evaluate and draw conclusions from the findings), and potentially Image Recognition (to interpret graphs and figures in academic papers).
>
> Given this variability, we follow the capability-based evaluation approach commonly used in previous studies and utilize our manually created taxonomy to distinguish the boundaries of different capabilities.
>
> **Clarification on multilingualism.** In terms of multilingualism, we maintain consistency with previous research [1-4] in treating English and other languages as distinct capabilities:
>
> For English:
>
> - We align with the Llama 3 paper [1], categorizing tasks such as knowledge-based QA and instruction following under general English.
> - English is treated as a distinct capability representing the foundational attributes of English-centric LLMs. Notably, it does not engage in cross-capability scenarios in our framework.
>
> For other languages, there is currently no consensus on whether they should be counted as one or more capabilities:
>
> - [1, 2] treat all non-English languages as a single capability.
> - [3] evaluates each language (e.g., Chinese, French, German) as an individual capability.
>
> In our work, we adopt the approach used in [4], selecting Spanish as a representative language to study cross capabilities. Notably, our cross-capability scenarios focus specifically on Spanish's interaction with other capabilities (e.g., Spanish & Reasoning, Spanish & Image Recognition), without making broader claims about the intersection of multilingualism with these capabilities.
>
> > **W1.3: On a related note, do we assume that the capabilities are equally different from each other?**
>
> Thank you for raising this insightful point about the varying "distances" between different capabilities. We do not assume that the capabilities are equally distinct from one another, and this is indeed an important consideration that influences our experimental design in two key ways:
>
> - For individual capabilities: We prioritize diversity among the selected capabilities.
>   - As you noted, some capabilities are relatively close in nature (e.g., different languages sharing similar linguistic structures)
>   - Others are distinctly different (e.g., coding and image recognition operating across different modalities)
>   - This diversity allows us to study capability interactions across varying degrees of similarity.
>
> - For cross capabilities: Our choice of cross-capability scenarios is guided by two main criteria:
>   - Coverage: Ensuring all individual capabilities are covered in cross-capability scenarios.
>   - Real-world relevance: Prioritizing combinations that frequently appear in actual user queries and applications.
>
> Notably, despite the varying "distances" between capabilities, our experiments consistently reveal the "Law of the Weakest Link" effect. This consistency across different capability combinations, regardless of their similarity or difference, suggests that our findings represent a fundamental characteristic of current LLMs rather than an artifact of specific capability selections.

---

> ### Author Response · Authors · 2024-11-22
> **Author Response (Part 3)**
>
> > **W2.1: How were the capabilities selected? What was left out? For instance, why not consider Long Context and Language?**
>
> Thank you for pointing out that we missed elaborating on this in the paper. The specific selection criteria are as follows:
>
> **Selection of Individual Capabilities.** We primarily follow the capability evaluations established for post-trained LLMs in the Llama 3 paper (see Table 16 in [1]), which represent capabilities of primary interest to the LLM community. Additionally, we include Image Recognition to reflect the growing multimodal capabilities of state-of-the-art LLMs. This ensures that our findings generalize across both language and vision modalities.
>
> **Selection of Cross Capabilities.** To select cross-capability scenarios, we ensure the coverage of all individual capabilities and analyze the frequency of capability pairs based on public real user queries [5, 6]. This analysis allows us to identify the seven most common and practically relevant cross-capability scenarios.
>
> Regarding your specific mention of Long Context & Language: while this is indeed a valid combination, our analysis of user queries shows that Long Context & Coding appears more frequently, leading to its inclusion in the CrossEval benchmark.
>
> > **W2.2: Also, how are the capabilities broken down into Levels 1 and 2? How exhaustive is this breakdown?**
>
> Thank you for asking about our taxonomy development process. The breakdown of capabilities into Levels 1 and 2 follows a multi-stage annotation process:
>
> **Annotation Team.** The annotation team consists of contributors with experience as LLM researchers or product managers, including some of the authors of this paper. The team's diverse expertise ensures that the taxonomy captures a wide range of real-world use cases.
>
> **L1 Annotation.** Each capability's L1 categories are derived from a large set of real user queries:
> - Three team members independently annotate the L1 categories by identifying common tasks in the queries
> - After the independent annotations, the team convenes to merge similar categories, address any omissions, and finalize the L1 taxonomy collaboratively
>
> **L2 Annotation.** Building on the agreed-upon L1 categories, the annotators independently propose L2 subcategories:
> - These finer-grained tasks are designed to provide additional specificity while maintaining consistency with the L1 framework.
> - A second round of discussions and revisions is conducted to refine the L2 categories and reach a consensus
>
> **Team Review.** Once the L1 and L2 categories are finalized, all taxonomies undergo a comprehensive review by the full team:
> - The team fine-tunes the taxonomy to ensure that all breakdowns adhere to a consistent granularity across capabilities and align with the [MECE principle](https://en.wikipedia.org/wiki/MECE_principle) (mutually exclusive and collectively exhaustive).
>
> **Annotation Effort.** The annotation effort involves more than ten main contributors and spans approximately one week.
>
> We hope this explanation clarifies the rigor of our taxonomy design and the exhaustiveness of the breakdown.
>
> > **W2.3: How were the examples for annotators generated (Appendix B.4)? Could these example prime the annotators? If these examples are not from the benchmark (as the text notes), what is their propose?**
>
> The difficulty level guidelines are developed by the same annotation team responsible for creating the taxonomy. Here are the details:
>
> **Source of the examples.** The examples are created by the team through several rounds of discussions. For each difficulty level of every capability, we provide at least 3 examples to illustrate the distinctions and establish clear expectations.
>
> **Purpose of the examples.** The primary purpose of these examples is to clarify the definitions of easy, medium, and hard prompts, which might otherwise lack sufficient clarity or intuitiveness. They serve as instructional tools to help professional annotators from our data vendor better understand our difficulty level criteria, enabling them to annotate prompts of varying difficulty accurately.
>
> **Exclusion from our benchmark.** As mentioned in the paper, these examples are not part of our benchmark dataset. Annotators are explicitly instructed not to copy these examples but to use them solely as a reference for understanding our distinctions and expectations for difficulty levels. This ensures that the examples serve as a guide without influencing or biasing the benchmark content.

---

> ### Author Response · Authors · 2024-11-22
> **Author Response (Part 4)**
>
> > **W2.4: Line 201: "selecting a leaf node from our taxonomy to categorize each prompt by capability". Where do these prompts come from?**
>
> We apologize for the confusion. The specific process for annotating prompts is as follows:
>
> We instruct annotators from the data vendor to randomly select a leaf node from the taxonomy, which determines the category to be annotated. Once the category is selected, the annotators create prompts according to the provided guidelines. In other words, we follow a **"category-first, prompt-second"** approach to ensure comprehensive coverage of our taxonomy.
>
> > **W2.5: How much coverage do these prompts provide of each capability?**
>
> As mentioned in Line 215, our 1,400 prompts cover all 76 Level-1 and 332 Level-2 categories.
>
> > **W2.6: There are 100 to 500 annotated prompts for each capability, but then why are the reviewers asked to select a precise 100 examples out of it?**
>
> Here’s how we address the selection and standardization of 100 prompts per capability:
>
> **Selection of 100 prompts.** Our reviewers follow a strict quality screening process to ensure all prompts align with the task categories in our taxonomy and meet the criteria outlined in the guidelines. If issues such as non-compliance, incomplete task coverage, or misalignment with the difficulty distribution are identified by our reviewers, the data vendor reannotates the problematic parts. After review and reannotation, the number of annotated prompts for each capability exceeds 100, with a unified quality control standard ensuring high-quality prompts across all capabilities.
>
> **Why limit to 100 prompts.** We limit 100 prompts for each capability for the following reasons:
> - *Consistency:* Standardizing to 100 prompts per capability ensures fair comparisons, preventing any capability from being over- or underrepresented due to variations in dataset size.
> - *Resource Efficiency:* Each prompt requires multiple model responses and expert annotations, which are resource-intensive. Limiting to 100 prompts maintains quality while managing costs and reduces the overall expense of evaluation on our benchmark.
>
> We hope this explanation clarifies our rationale and highlights the efforts taken to ensure high-quality and consistent annotations.
>
> > **W3.1: Why is “Brainstorming about Local and Current Things” a part of the Tool Use? Could you provide examples of tools that perform brainstorming?**
>
> We apologize for any confusion. The full name of this category is "Recommendations / Brainstorming about Local and Current Things." This category involves leveraging tools to gather information about "Local and Current Things" to complete tasks related to "Recommendations" or "Brainstorming." For instance, tools such as web browsers can be employed to search for recent events, local services, or trends to generate relevant recommendations or brainstorm ideas.
>
> ---
>
> **Please note that all detailed explanations regarding the issues you mentioned have been updated in the new version of the manuscript.** If you still have any concerns, please let us know. We would be happy to address them further and update our paper accordingly.
>
> ---
>
> **References:**
>
> [1] Llama Team. The Llama 3 Herd of Models.
>
> [2] Kim et al. The BiGGen Bench: A Principled Benchmark for Fine-grained Evaluation of Language Models with Language Models
>
> [3] Chatbot Arena LLM Leaderboard. https://lmarena.ai.
>
> [4] SEAL LLM Leaderboards. https://scale.com/leaderboard.
>
> [5] Zheng et al. LMSYS-Chat-1M: A Large-Scale Real-World LLM Conversation Dataset. ICLR 2024.
>
> [6] Zhao et al. WildChat: 1M ChatGPT Interaction Logs in the Wild. ICLR 2024.

---

> > ### Author Response · Authors · 2024-11-30
> > **Looking forward to the discussion**
> >
> > Dear Reviewer 7cFP,
> >
> > Thank you for dedicating your time and effort to reviewing our work. We understand that your schedule may be quite busy and truly appreciate your thoughtful evaluation. As the author-reviewer discussion phase approaches its end, we kindly request your feedback on whether our response has effectively addressed your concerns and if there are any additional questions or points you would like to discuss.
> >
> > Looking forward to further discussion and your valuable insights.
> >
> > Best regards,
> >
> > The Authors

---

> > > ### Author Response · Authors · 2024-12-03
> > > **Kind Reminder on Author-Reviewer Discussion**
> > >
> > > Dear Reviewer 7cFP,
> > >
> > > As the discussion phase concludes in a few hours, we’d like to gently remind you to let us know if you have any remaining questions or concerns. We’re more than happy to provide further clarification if needed.
> > >
> > > Thank you for your time and insights!
> > >
> > > Best regards,
> > >
> > > The Authors

---

### Author Response · Authors · 2024-11-22
**General Response**

Dear Reviewers,

We are deeply grateful for your insightful feedback and valuable suggestions. Your comprehensive reviews have provided us with essential guidance to enhance our work. We would like to start by expressing our appreciation for the positive recognition of the strengths of our study, including:

- **Research Topic:** The paper’s focus on cross-capability evaluation and investigation is both interesting and important. (`7cFP`, `A1B6`, `Gjoy`)
- **Taxonomy Contribution:** The annotated taxonomies for the 14 capabilities of LLMs are meaningful (`A1B6`, `zpqw`)
- **Prompt Annotation:** The methodology for prompt collection and annotation is comprehensive and reliable (`7cFP`, `Gjoy`, `zpqw`)
- **Evaluation Protocol:** The model-based evaluation protocol is meticulously designed (`7cFP`)
- **Benchmark Contribution:** The CrossEval benchmark is a good contribution to the field (`zpqw`)
- **Key Insights:** The insights and findings about the "Law of the Weakest Link" are particularly valuable (`Gjoy`, `zpqw`)
- **Paper Writing:** The paper is well-written and easy to follow (`7cFP`, `A1B6`, `zpqw`)

We have responded individually to each reviewer’s questions and incorporated their suggestions into our revised manuscript. To ensure clarity, **all changes are highlighted in blue.** Below is a summary of the key updates:

- Included selection criteria for individual and cross capabilities (Section 2.1 and 2.2)
- Provided more details about our taxonomy construction (Appendix A.3)
- Added principles and definitions for each capability (Introduction, Section 2.1, Appendix B.1)
- Clarified the distinction between capability and skill (Appendix B.2)
- Provided additional clarification on multilingualism (Appendix B.3)
- Added the discussions on the distance between capabilities (Appendix B.4)
- Clarified why different capabilities are comparable (Appendix B.5)
- Discussed why constructing the benchmark from scratch is necessary (Appendix C.1)
- Included selection process and criteria for Prompts (Appendix C.2)
- Refined the descriptions for our annotation procedure (Section 3.1, Appendix C.6)

We sincerely thank you again for your contributions to improving our work. If there are any additional concerns or questions, we are fully prepared to address them.

---

### Meta-Review · Area_Chair_gkJb · 2024-12-23

**Metareview:**

The paper introduces a dataset for LLM evaluation of "cross capabilities", the intersection of multiple (in this paper, two) abilities. All the reviewers appreciated the paper and, despite they highlighted a number of points for improvement, they were mildly positive or positive in terms of overall score. As a consequence, I recommend acceptance.

**Additional Comments On Reviewer Discussion:**

The authors put a significant effort during the rebuttal period and persuaded some of the reviewers to increase their overall score.

---

### Decision · Program_Chairs · 2025-01-22

Accept (Poster)